# Conformal Robustness Control: A New Strategy for Robust Decision

**Yang Hu[1], Jieren Tan[2], Changliang Zou[2], Yajie Bao[2]* & Haojie Ren[1]***

[1]School of Mathematical Sciences, Shanghai Jiao Tong University
[2]School of Statistics and Data Sciences, LPMC, KLMDASR and LEBPS, Nankai University
`{hu_yang,haojieren}@sjtu.edu.cn`
`tanjr@mail.nankai.edu.cn`
`{yajiebao,zoucl}@nankai.edu.cn`

## Abstract

Robust decision-making is crucial in numerous risk-sensitive applications where outcomes are uncertain and the cost of failure is high. Conditional Robust Optimization (CRO) offers a framework for such tasks by constructing prediction sets for the outcome that satisfy predefined coverage requirements and then making decisions based on these sets. Many existing approaches leverage conformal prediction to build prediction sets with guaranteed coverage for CRO. However, since coverage is a *sufficient but not necessary* condition for robustness, enforcing such constraints often leads to overly conservative decisions. To overcome this limitation, we propose a novel framework named Conformal Robustness Control (CRC), that directly optimizes the prediction set construction under explicit robustness constraints, thereby enabling more efficient decisions without compromising robustness. We develop efficient algorithms to solve the CRC optimization problem, and also provide theoretical guarantees on both robustness and optimality. Empirical results show that CRC consistently yields more effective decisions than existing baselines while still meeting the target robustness level.

## 1 Introduction

In many real-world applications, it is crucial for decision-makers to account for operational risks to avoid irreversible consequences. For example, portfolio management (Markowitz, 1952) aims to maximize returns while navigating the trade-off with risk tolerance. Similar risk-sensitive decision-making challenges are also evident in fields such as medical diagnosis (Kiyani et al., 2025) and transportation planning (Patel et al., 2024).

Consider a scenario where we observe an input $X$, but the corresponding outcome $Y$ is unknown. The decision-maker needs to choose a decision $z(X)$ based on the input $X$ such that the incurred decision loss $\phi(Y, z(X))$ does not exceed a certain *risk certificate* $r(X)$ with high probability. Formally, the $(1 - \alpha)$-level robustness requirement is given by

$$\mathbb{P}\{\phi(Y, z(X)) \leq r(X)\} \geq 1 - \alpha. \tag{1}$$

At the same time, the decision-maker seeks to minimize $r(X)$ to improve efficiency and reduce potential worst-case losses.

Over the years, Conditional Robust Optimization (CRO), introduced by Chenreddy et al. (2022), has become a widely adopted and effective framework for robust decision-making. As an extension of classical robust optimization (Ben-Tal et al., 2009), CRO incorporates covariate information to enhance decision quality, enabling more precise and context-aware responses in complex tasks. In the CRO framework, decisions are derived from a minmax optimization problem using a prediction set $\mathcal{U}(X)$, formulated as $z_{\mathcal{U}}(X) := \arg\min_{z \in \mathcal{Z}} \max_{y \in \mathcal{U}(X)} \phi(y, z)$. By designing $\mathcal{U}(X)$ with a regular structure, such as a box or an ellipse, the resulting minmax problem remains convex and can be solved efficiently in polynomial time. The corresponding risk certificate value is defined

---

*Corresponding authors

as $r_{\mathcal{U}}(X) := \max_{y \in \mathcal{U}(X)} \phi(y, z_{\mathcal{U}}(X))$. To meet the robustness requirement (1), CRO enforces a coverage condition on the prediction set, that is

$$\mathbb{P}\{Y \in \mathcal{U}(X)\} \geq 1 - \alpha. \qquad (2)$$

Recently, Johnstone & Cox (2021) and Sun et al. (2023) first employed the conformal prediction (Vovk et al., 2005) to construct the prediction set $\mathcal{U}(X)$ from historical labeled data with the target coverage level $1 - \alpha$; then substituted it into the minmax problem to make the final decision. It can be observed that the coverage property (2) is a sufficient condition for achieving the final robustness target (1). Hence, the two-step procedure above provides a statistically valid robustness guarantee for the subsequent decisions.

However, as noted by Ben-Tal et al. (2009), controlling robustness via this sufficient condition often results in suboptimal and overly conservative decisions. In this paper, we introduce Conformal Robustness Control (CRC), a new strategy to alleviate the conservativeness of the existing CRO framework and to enable more efficient robust decisions. Our contributions are summarized as follows.

(1) Unlike conventional CRO methods that enforce coverage constraint on prediction sets, CRC directly minimizes the expected risk certificate under explicit robustness constraint, significantly improving decision efficiency. The CRC procedure is amenable to efficient gradient-based optimization algorithms that minimize an empirical loss using labeled data.

(2) We establish non-asymptotic theoretical guarantees on both the robustness and the optimality gap of the resulting decisions. For a given test data point, we further develop a sample-splitting calibration procedure to endow the optimized prediction set with finite-sample robustness guarantees.

(3) Through extensive experiments on both synthetic data and real-world applications, the proposed CRC consistently outperforms baseline methods across key metrics.

Figure 1 compares the conventional CRO framework with our proposed method CRC, at the nominal robustness level $1 - \alpha = 90\%$. The brown circular regions in the right panel represent prediction sets $\mathcal{U}(X)$ satisfying a 90% coverage constraint and a 90% robustness constraint, respectively. The CRO decision attains a robustness level of 98%, which is significantly higher than the nominal requirement. This leads to a higher risk certificate and decision loss compared to the proposed CRC.

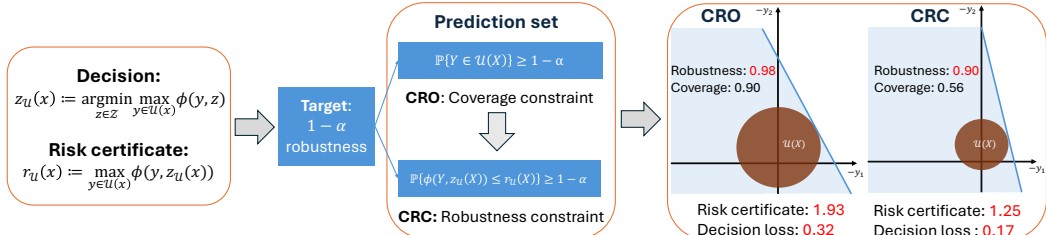

Figure 1: Comparison of CRO and our method CRC. Portfolio optimization problem with $\phi(y, z) = -y^\top z$, $\mathcal{Z} = \{z \in \mathbb{R}^2 : z_1 + z_2 = 1, z_1, z_2 \geq 0\}$, and $\alpha = 0.1$. Blue lines show CRO solutions for the brown circular prediction sets. The shaded blue regions indicate where the loss $\phi(y, z(X))$ is below the risk certificate $r(X)$. The prediction set in CRO achieves exact 90% coverage, with $r(X) = 1.93$. In contrast, CRC meets the 90% robustness requirement, yielding a more efficient decision with $r(X) = 1.25$.

## 2 RELATED WORKS

Robust optimization is a well-established method for decision-making under uncertainty. Early work focused on approximating Value at Risk (VaR) by designing deterministic prediction sets to induce robustness (Ghaoui et al., 2003; Natarajan et al., 2008; Bertsimas et al., 2018). Later studies,

such as Shang et al. (2017); Bertsimas et al. (2018); Hong et al. (2021), have proposed data-driven prediction sets. With the growing size of data, Chenreddy et al. (2022) explored how covariate information could be leveraged to develop more effective prediction sets, leading to the introduction of the Conformal Robust Optimization (CRO) framework. Subsequent works by Johnstone & Cox (2021); Patel et al. (2024); Sun et al. (2023) incorporated conformal prediction methods to construct prediction sets that satisfy coverage conditions, thereby providing finite-sample robustness guarantees for CRO. Kiyani et al. (2025) derived the explicit form of the optimal prediction set that has the minimum risk certificate under the coverage constraint. However, the construction relies on minimizing the VaR function, which often also leads to intractable formulations if the decision space is continuous (Uryasev & Rockafellar, 2001). In addition, Wang et al. (2023) also considered optimizing the prediction sets in a robust optimization problem, but relaxing the robustness constraint through the conditional Value at Risk transformation (Rockafellar & Uryasev, 2002). Compared to existing work, we impose the exact robustness constraint rather than a coverage constraint on the prediction set, thereby enhancing the generation of more effective decisions.

Conformal prediction is a widely used method for uncertainty quantification, notable for its model-agnostic and distribution-free properties (Vovk et al., 2005; Lei et al., 2018; Angelopoulos et al., 2024a). In predictive inference tasks, the efficiency measure of conformal prediction sets is the size or volume. Recent research has increasingly focused on improving the efficiency of these prediction sets. Several studies, such as Sadinle et al. (2019), Bai et al. (2022), Stutz et al. (2022), and Kiyani et al. (2024b) have formulated constrained optimization problems that minimize the size of prediction sets subject to coverage constraints. In addition, Yang & Kuchibhotla (2025) introduced a sample-splitting approach to select models yielding the smallest prediction sets, followed by constructing split conformal prediction sets (Vovk et al., 2005; Papadopoulos et al., 2002). Differently, Liang et al. (2024) proposed a method that avoids sample splitting while maintaining finite-sample coverage during model selection. In terms of decision efficiency, since the performance of decisions varies significantly with different conformal prediction sets, Chenreddy & Delage (2024) and Yeh et al. (2024) proposed end-to-end learning methods that train the conformal prediction sets by directly minimizing downstream expected decision risk. Moreover, Bao et al. (2025) developed new frameworks for prediction set selection in the CRO problem, which could keep finite-sample robustness control while avoiding sample splitting.

# 3 PREDICTION SET OPTIMIZATION WITH ROBUSTNESS CONTROL

## 3.1 PROBLEM SETUP

Let $\mathcal{X}$ be the covariate space, and $\mathcal{Y}$ be the label space. The primary goal of robust decision is to find a decision policy $z(\cdot) : \mathcal{X} \to \mathcal{Z}$ and a risk certificate function $r(\cdot) : \mathcal{X} \to \mathbb{R}$ that minimizes $\mathbb{E}[r(X)]$ subject to the robustness constraint in (1). It is consistent with the Risk Averse Decision Policy Optimization (RA-DPO) problem defined by Kiyani et al. (2025):

$$\min_{z(\cdot),r(\cdot)} \mathbb{E}[r(X)] \quad \text{s.t.} \quad \mathbb{P}\{\phi(Y, z(X)) \leq r(X)\} \geq 1 - \alpha. \tag{3}$$

However, directly optimizing over arbitrary forms of $z(\cdot)$ and $r(\cdot)$ is generally difficult. The CRO framework provides a flexible alternative by introducing a prediction set $\mathcal{U}(\cdot)$ that maps each covariate $x \in \mathcal{X}$ to a subset of the label space $\mathcal{Y}$, which relates to both the decision and the associated risk certificate. Specifically, for $x \in \mathcal{X}$,

$$z_{\mathcal{U}}(x) := \arg\min_{z \in \mathcal{Z}} \max_{y \in \mathcal{U}(x)} \phi(y, z), \quad r_{\mathcal{U}}(x) = \max_{y \in \mathcal{U}(x)} \phi(y, z_{\mathcal{U}}(x)).$$

To identify the optimal decision policy and risk certificate, it is natural to minimize the expected risk certificate under the robustness constraint:

$$\min_{\mathcal{U}(\cdot):\mathcal{X} \to 2^{\mathcal{Y}}} \mathbb{E}[r_{\mathcal{U}}(X)] \quad \text{s.t.} \quad \mathbb{P}\{\phi(Y, z_{\mathcal{U}}(X)) \leq r_{\mathcal{U}}(X)\} \geq 1 - \alpha. \tag{4}$$

The next theorem shows the equivalence between RA-DPO in (3) and the problem (4), which also means that optimizing the prediction sets will not result in a suboptimal risk certificate.

**Theorem 3.1.** *Let $z^{RA\text{-}DPO}(\cdot), r^{RA\text{-}DPO}(\cdot)$ be the optimal solution of RA-DPO in* (3)*, and let $\mathcal{U}^*$ be the optimal solution of* (4)*. It holds that $\mathbb{E}[r^{RA\text{-}DPO}(X)] = \mathbb{E}[r_{\mathcal{U}^*}(X)]$, which means problems* (3) *and* (4) *are equivalent in minimizing the expected risk certificate while maintaining robustness control.*

We defer the proof of Theorem 3.1 to Appendix B.2. The prior work of Kiyani et al. (2025) also derived a formulation equivalent to RA-DPO in Eq. (3), termed Risk Averse Conformal Prediction Optimization (RA-CPO). This formulation optimizes the expected risk certificate over all possible prediction sets subject to a coverage constraint, that is, replacing the constraint in (4) with (2). To construct the "optimal" prediction set that solves RA-CPO, Kiyani et al. (2025) proposed a method based on the minimizer and minimum value of a contextual VaR problem: $z^*(x) = \arg\min_{z \in \mathcal{Z}} \mathrm{VaR}_{1-\alpha}(\phi(Y,z)|X=x)$ and $r^*(x) = \min_{z \in \mathcal{Z}} \mathrm{VaR}_{1-\alpha}(\phi(Y,z)|X=x)$. Here, $\mathrm{VaR}(\cdot|X=x)$ denotes the conditional $1 - \alpha$ population quantile given the covariate $X = x$. In the case of classification with a finite decision space, Kiyani et al. (2024a) approximated the optimal prediction set by first estimating the conditional distribution of $Y \mid X$, then the associated VaR problem can be solved by traversal. However, this approach does not extend to continuous decision spaces $\mathcal{Z}$, where the VaR problem generally becomes intractable (Uryasev & Rockafellar, 2001).

To address this limit, we consider solving problem (4) over the parametrized prediction set $\mathcal{U}_\theta(\cdot)$, where $\theta \in \Theta$ refers to the model parameters. In regression settings with $\mathcal{X} = \mathbb{R}^p, \mathcal{Y} = \mathbb{R}^q$, two commonly used types of prediction sets are *box* and *ellipsoidal* sets (Johansson et al., 2017; Sun et al., 2023). Their parametrized forms are can be defined as follows.

- **Box prediction set.** A box-shaped prediction set is constructed by componentwise lower and upper bounds for the response vector. Let $h_\theta^{\mathrm{lo}}(\cdot) : \mathbb{R}^p \to \mathbb{R}^q$ and $h_\theta^{\mathrm{hi}}(\cdot) : \mathbb{R}^p \to \mathbb{R}^q$ be models with parameters $\theta \in \Theta$, then

$$\mathcal{U}_\theta(x) = \left\{ y \in \mathbb{R}^q : h_\theta^{\mathrm{lo}}(x) \leq y \leq h_\theta^{\mathrm{hi}}(x) \right\}.$$

- **Ellipsoidal prediction set.** Unlike box sets, ellipsoidal prediction sets account for correlations among components of the response vector. Let $\mu_\theta(\cdot) : \mathbb{R}^p \to \mathbb{R}^q$ and $\Sigma_\theta(\cdot) : \mathbb{R}^p \to \mathbb{R}^{q \times q}$ denote the mean and covariance model with parameters $\theta \in \Theta$, then

$$\mathcal{U}_\theta(x) = \left\{ y \in \mathbb{R}^q : (y - \mu_\theta(x))^\top \Sigma_\theta^{-1}(x) (y - \mu_\theta(x)) \leq 1 \right\}.$$

In Appendix E.6, we also provide the example of a parametrized polyhedral set based on the definition in Bärmann et al. (2016).

For a parametrized prediction set $\mathcal{U}_\theta(\cdot)$, we denote the corresponding decision policy and risk certificate functions as $z_\theta(\cdot) \equiv z_{\mathcal{U}_\theta}(\cdot)$ and $r_\theta(\cdot) \equiv r_{\mathcal{U}_\theta}(\cdot)$ for short. We then consider the parameterized version of problem (4):

$$\min_{\theta \in \Theta} \mathbb{E}[r_\theta(X)] \quad \text{s.t.} \quad \mathbb{P}\left\{ \phi(Y, z_\theta(X)) \leq r_\theta(X) \right\} \geq 1 - \alpha. \tag{5}$$

Even though the parametrized optimization can also be applied to the RA-CPO framework, we show that our proposed problem (5) yields a lower risk certificate in Appendix B.3, which further confirms the benefit of robustness constraint over the coverage constraint. In the following subsections, we investigate the optimization problem (5) based on the collected labeled data and provide theoretical results for the robustness and optimality guarantees. Moreover, we also developed a differential algorithm to solve the optimization problem for the continuous decision space.

## 3.2 EMPIRICAL OPTIMIZATION WITH CONFORMAL ROBUSTNESS CONTROL

Suppose we have collected an i.i.d. labeled dataset $\mathcal{D}_n = \{(X_i, Y_i)\}_{i=1}^n$ drawn from some distribution $P$. We first optimize the prediction set by addressing an empirical version of the problem (5), and then apply this prediction set for decision-making. By approximating both the objective and the constraint in (5) via sample averaging, we obtain the following empirical counterpart:

$$\hat{\theta} = \arg\min_{\theta \in \Theta} \frac{1}{n} \sum_{i=1}^n r_\theta(X_i) \quad \text{s.t.} \quad \frac{1}{n} \sum_{i=1}^n \mathbb{1}\left\{ \phi\left(Y_i, z_\theta(X_i)\right) \leq r_\theta(X_i) \right\} \geq 1 - \alpha. \tag{6}$$

To distinguish from coverage control methods and to emphasize the explicit robustness constraint, we refer to this procedure as *Conformal Robustness Control* (CRC).

A natural approach to solving problem (6) is to consider its dual formulation. Define the Lagrangian function as $L(\lambda; \theta) := f(\theta) + \lambda g(\theta)$, where $\lambda \geq 0$ is the Lagrange multiplier, $f(\theta) = $

$\frac{1}{n}\sum_{i=1}^{n} r_\theta(X_i)$ and $g(\theta) = 1 - \alpha - \frac{1}{n}\sum_{i=1}^{n} \mathbb{1}\{\phi(Y_i, z_\theta(X_i)) \leq r_\theta(X_i)\}$. The function $f(\theta)$ is typically differentiable if the CRO problem for $\mathcal{U}_\theta(x)$ can be reformulated into a convex programming. In such cases, its gradient can be computed using existing implicit differential tools, see Amos & Kolter (2017) and Agrawal et al. (2019). In contrast, the term $g(\theta)$ is non-smooth due to the indicator. To enable gradient-based optimization, we approximate the indicator with a smooth surrogate $\tilde{\mathbb{1}}\{a \leq b\} = \frac{1}{2}(1 + \text{erf}(\frac{b-a}{\sqrt{2}\sigma}))$, where $\text{erf}(x) = \frac{2}{\sqrt{\pi}}\int_0^x e^{-t^2} dt$ is the Gaussian error function and $\sigma > 0$ controls the smoothness. Replacing the indicator in $g(\theta)$ with this surrogate yields a smoothed constraint function $\tilde{g}(\theta)$. Similar smoothing techniques have been employed in the optimization problem of conformal prediction (Bai et al., 2022; Kiyani et al., 2024b; Wu et al., 2025). The resulting smoothed dual problem is given by: $\min_{\theta \in \Theta} \max_{\lambda \geq 0} \widetilde{L}(\lambda; \theta)$, where $\widetilde{L}(\lambda; \theta) = f(\theta) + \lambda\tilde{g}(\theta)$. This smooth approximation enables numerical solution via an alternating gradient descent algorithm. We refer to Davis et al. (2020) and Bolte et al. (2021) for the convergence analysis of similar optimization problems. Implementation details are summarized in Algorithm 1.

---

**Algorithm 1** Prediction Set Optimization with CRC

---

1: **Input**: Loss function $\phi$, robustness level $1 - \alpha$, labeled dataset $\mathcal{D}_n = \{(X_i, Y_i)\}_{i=1}^n$, parametrized set $\mathcal{U}_\theta(\cdot)$ with $\theta \in \Theta$, smooth surrogate function $\tilde{\mathbb{1}}$, learning rate $\eta > 0$.
2: Initialize $\theta \leftarrow \theta_0$ and $\lambda = 0$.
3: Compute $r_\theta(X_i)$ and $z_\theta(X_i)$ for $i \in [n]$.
4: Define empirical objective $f(\theta)$ and set smooth constraint $\tilde{g}(\theta)$.
5: Form the smoothed Lagrange multiplier $\widetilde{L}(\lambda; \theta) \leftarrow f(\theta) + \lambda\tilde{g}(\theta)$.
6: **while** no converged **do**
7:     Perform a few steps of gradient descent on $\theta$ to minimize $\tilde{L}(\lambda; \theta)$.
8:     Compute $\tilde{g}(\theta)$ and perform projected gradient asent $\lambda \leftarrow \max\{0, \lambda + \eta\tilde{g}(\theta)\}$.
9: **end while**
10: $\hat{\theta} \leftarrow \theta$.
11: **Output**: Prediction set $\mathcal{U}_{\hat{\theta}}(\cdot)$.

---

**Remark 3.1.** *In learning problems, Angelopoulos et al. (2024b) proposed the framework named conformal risk control by extending the miscoverage risk to general monotone risk functions. The robustness constraint can be regarded as a special risk, whereas it is not monotone in the model parameter. In addition, the objective function in Angelopoulos et al. (2024b) is the threshold parameter of prediction sets, but we consider the risk certificate function $r(\cdot)$, which is more complex.*

### 3.3 THEORETICAL RESULTS

This section presents the theoretical guarantees for the solution to problem (6). The analysis for the smoothed variant (Algorithm 1), being conceptually analogous, are deferred to Appendix D.2. We equip the parameter space $\Theta$ with the supremum norm and state the underlying assumptions.

**Condition 3.1.** *Loss function $\phi$ is $L_\phi$-Lipschitz in decision $z$ for any $y \in \mathcal{Y}$. For any $x \in \mathcal{X}$, the decision $z_\theta(x)$ is $L_z$-Lipschitz in $\theta$. The risk certificate $r_\theta(x)$ is $L_r$-Lipschitz in $\theta \in \Theta$, and uniformly bounded by a positive constant $B_r > 0$ for any $x \in \mathcal{X}$ and $\theta \in \Theta$.*

These regularity conditions are mild and typically satisfied in practice. For example, in portfolio optimization with the loss function $\phi(y, z) = -y^\top z$, the Lipschitz condition holds if $\mathcal{Y}$ is bounded. For decision function $z_\theta$ and risk certificate function $r_\theta$, if the CRO problem for prediction set $\mathcal{U}_\theta$ can be transformed into a smooth convex optimization problem, then the Lipschitz property can be derived from the KKT conditions and the implicit function theorem, see Bolte et al. (2021) and Amos & Kolter (2017). The next assumption introduces a mild distributional assumption.

**Condition 3.2.** *Let $V_\theta(X, Y) = \phi(Y, z_\theta(X)) - r_\theta(X)$ for data $(X, Y) \sim P$. Suppose that for all $\theta \in \Theta$ the density of $V_\theta(X, Y)$ is uniformly bounded by a constant $\rho_0 > 0$.*

The bounded density condition is often needed for concentration guarantees in the conformal prediction literature (Kiyani et al., 2024b; Jung et al., 2023; Lei & Wasserman, 2014).

**Definition 3.1** (**Covering number**). *Let $\Theta$ be a parameter space with the supremum norm $\|\cdot\|_\infty$. Given any $\epsilon > 0$, the subset $\Theta_\epsilon \subseteq \Theta$ is called an $\epsilon$-covering of $\Theta$ if for every $\theta \in \Theta$, there exists*

some $\theta_\epsilon \in \Theta_\epsilon$ such that $\|\theta - \theta_\epsilon\|_\infty < \epsilon$. The covering number $\mathcal{N}(\Theta, \|\cdot\|_\infty, \epsilon)$ is the smallest cardinality of any $\epsilon$-covering of $\Theta$.

Covering numbers quantify the complexity of a function class and are a fundamental tool in statistical learning theory and convergence analysis (Van Der Vaart & Wellner, 1996). The next two theorems provide a non-asymptotic characterization of the robustness and expected risk certificate value for CRC.

**Theorem 3.2** (**Robustness gap**). *Let $\hat{\theta}$ be the solution to optimization problem (6). Under Conditions 3.1 and 3.2, for any independent data $(X, Y) \sim P$, conditioning on the labeled data $\mathcal{D}_n$, the following inequality holds: with probability at least $1 - n^{-1}$,*

$$\mathbb{P}\left\{\phi\left(Y, z_{\hat{\theta}}(X)\right) \leq r_{\hat{\theta}}(X) \mid \mathcal{D}_n\right\} \geq 1 - \alpha - \Delta_n,$$

*where the robustness gap* $\Delta_n = 5\sqrt{\frac{\log(2\mathcal{N}(\Theta, \|\cdot\|_\infty, n^{-1})) + \log n}{2n}} + \frac{4(L_\phi L_z + L_r)\rho_0}{n}$.

**Theorem 3.3** (**Risk certificate optimality**). *Let $\theta^*_{\Delta_n}$ denote the optimal solution of problem (5) at the robustness level $1 - \alpha + \Delta_n$. Under the same conditions as Theorem 3.2, conditioning on $\mathcal{D}_n$, with probability at least $1 - 2n^{-1}$,*

$$\mathbb{E}\left[r_{\hat{\theta}}(X) - r_{\theta^*_{\Delta_n}}(X) \mid \mathcal{D}_n\right] \leq 4B_r\sqrt{\frac{\log(2\mathcal{N}(\Theta, \|\cdot\|_\infty, n^{-1})) + \log n}{2n}} + \frac{4L_r}{n}.$$

For a finite-dimensional parameter space $\Theta$ of dimension $d$, the covering number scales approximately as $\mathcal{N}(\Theta, \|\cdot\|_\infty, n^{-1}) \asymp n^d$, so both the robustness gap and the expected risk certificate converge to zero at rate $O(\sqrt{d \log n / n})$. In Appendix D, we provide more comprehensive theoretical results, such as in the setting where the function class has a finite VC dimension.

**Remark 3.2.** *It is worth noting that $\theta^*_{\Delta_n}$ denotes the optimal model under a slightly relaxed robustness level $1 - \alpha + \Delta_n$, rather than the exact level $1 - \alpha$. This relaxation is introduced to ensure that $\theta^*_{\Delta_n}$ is feasible to the problem (6) with high probability, thereby guaranteeing that the empirical risk certificate of $\hat{\theta}$ is less than that of $\theta^*_{\Delta_n}$ with high probability. Finally, leveraging relevant theories of empirical process, we can establish the bounds in Theorems 3.2 and 3.3. Let $\theta^*$ be the solution to the problem (4). If additional assumptions are imposed regarding the $\|\theta^*_{\Delta_n} - \theta^*\|_\infty$, such a relaxation may no longer be needed.*

## 4 TEST-TIME DECISION WITH FINITE-SAMPLE ROBUSTNESS CONTROL

In this section, we turn to the practical task of making decisions at a *test point* $X_{n+1}$ with unknown label $Y_{n+1}$. A straightforward approach is to output the decision $z_{\mathcal{U}_{\hat{\theta}}}(X_{n+1})$, where $\hat{\theta}$ is solution to the problem (6). Theorem 3.2 shows that the robustness of the decision $z_{\mathcal{U}_{\hat{\theta}}}(X_{n+1})$ converges to the target level asymptotically. To achieve *finite-sample* robustness control for the decision of the specific test point, we further calibrate the prediction set obtained from Algorithm 1 using both the test data $X_{n+1}$ and the labeled data $\{(X_i, Y_i)\}_{i=1}^n$.

Specifically, we split the labeled dataset $\mathcal{D}_n$ into a training set $\mathcal{D}_{\text{train}} = \{(X_i, Y_i)\}_{i=1}^{n_0}$ and a calibration set $\mathcal{D}_{\text{cal}} = \{(X_i, Y_i)\}_{i=n_0+1}^n$, where $n_0 < n$. We first obtain the optimized prediction set $\mathcal{U}_{\hat{\theta}_0}(\cdot)$ using only $\mathcal{D}_{\text{train}}$ in Algorithm 1. Next, we apply full conformal prediction (Vovk et al., 2005; Lei et al., 2018) to calibrate the prediction set $\mathcal{U}_{\hat{\theta}_0}(\cdot)$ based on $\mathcal{D}_{\text{cal}}$ and $X_{n+1}$.

Calibrating the entire parameters $\theta$ is computationally expensive and often unnecessary. Instead, we can adjust the prediction set $\mathcal{U}_{\hat{\theta}_0}(\cdot)$ by tuning a single radius parameter $t \in \mathbb{R}^+$, which controls the size of the set and provides an efficient way of model calibration. Following the framework of nested prediction set in Gupta et al. (2022), we call the family $\{\mathcal{U}_{\theta,t}(x)\}_{t \in \mathbb{R}^+}$ *nested sets* if $t_1 \leq t_2$ implies that $\mathcal{U}_{\theta,t_1}(x) \subseteq \mathcal{U}_{\theta,t_2}(x)$ for any $x \in \mathcal{X}$. For the two examples of prediction sets in Section 3.1, the nested versions are given as follows.

- Nested parametrized box set:

$$\mathcal{U}_{\theta,t}(x) = \left\{y \in \mathbb{R}^q : h_\theta^{\text{lo}}(x) - t \leq y \leq h_\theta^{\text{hi}}(x) + t\right\};$$

- Nested parametrized ellipsoidal set:
$$\mathcal{U}_{\theta,t}(x) = \{y \in \mathbb{R}^q : (y - \mu_\theta(x))^\top \Sigma_\theta^{-1}(x) (y - \mu_\theta(x)) \le t\}.$$

Let $y \in \mathcal{Y}$ be a *hypothesized value* for the test label $Y_{n+1}$, and denote the augmented calibration set as $\{(X_i, Y_i^y)\}_{i=n_0+1}^{n+1}$, where $Y_i^y = Y_i$ for $n_0 + 1 \le i \le n$ and $Y_{n+1}^y = y$. Given the prediction set $\mathcal{U}_{\hat{\theta}_0,t}(\cdot)$, the hypothesized radius threshold is computed by

$$\hat{t}^y = \min\left\{t \in \mathbb{R}^+ : \frac{1}{n - n_0 + 1} \sum_{i=n_0+1}^{n+1} \mathbb{1}\left\{\phi\big(Y_i^y, z_{\hat{\theta}_0,t}(X_i)\big) \le r_{\hat{\theta}_0,t}(X_i)\right\} \ge 1 - \alpha\right\}, \quad (7)$$

where $z_{\theta,t}(x) := \arg\min_{z \in \mathcal{Z}} \max_{c \in \mathcal{U}_{\theta,t}(x)} \phi(c, z)$ and $r_{\theta,t}(x) := \max_{c \in \mathcal{U}_{\theta,t}(x)} \phi(c, z_{\theta,t}(x))$. Then the calibrated prediction set is given by

$$\mathcal{U}_{\text{Cal}}(X_{n+1}) = \left\{y \in \mathcal{Y} : \phi\big(y, z_{\hat{\theta}_0,\hat{t}^y}(X_{n+1})\big) \le r_{\hat{\theta}_0,\hat{t}^y}(X_{n+1})\right\}.$$

Finally, the decision for test point is made by $z_{\mathcal{U}_{\text{Cal}}}(X_{n+1})$. We name the procedure above as Calibrated CRC (Cal-CRC), and summarize it in Algorithm 2.

---

**Algorithm 2** Cal-CRC

---

1: **Input**: Same as Algorithm 1, size of training set $n_0$, and test point $X_{n+1}$.
2: **Sample splitting**: $\mathcal{D}_{\text{train}} = \{(X_i, Y_i)\}_{i=1}^{n_0}$ and $\mathcal{D}_{\text{cal}} = \{(X_i, Y_i)\}_{i=n_0+1}^{n}$.
3: **Training**: Obtain the prediction set $\mathcal{U}_{\hat{\theta}_0}(\cdot)$ by running Algorithm 1 on $\mathcal{D}_{\text{train}}$.
4: **Calibration**: $\mathcal{U}_{\text{Cal}}(X_{n+1}) \leftarrow \emptyset$.
5: **for** $y \in \mathcal{Y}$ **do**
6:      Define $\{(X_i, Y_i^y)\}_{i=n_0+1}^{n+1}$, where $Y_i^y = Y_i$ for $n_0 + 1 \le i \le n$ and $Y_{n+1}^y = y$.
7:      Calculate the hypothesized threshold $\hat{t}^y$ via (7).
8:      **if** $\phi\big(y, z_{\hat{\theta}_0,\hat{t}^y}(X_{n+1})\big) \le r_{\hat{\theta}_0,\hat{t}^y}(X_{n+1})$ **then**
9:          $\mathcal{U}_{\text{Cal}}(X_{n+1}) \leftarrow \mathcal{U}_{\text{Cal}}(X_{n+1}) \cup \{y\}$.
10:      **end if**
11: **end for**
12: Make the decision: $z_{\mathcal{U}_{\text{Cal}}}(X_{n+1}) \leftarrow \arg\min_{z \in \mathcal{Z}} \max_{y \in \mathcal{U}_{\text{Cal}}(X_{n+1})} \phi(y, z)$.
13: **Output**: the decision $z_{\mathcal{U}_{\text{Cal}}}(X_{n+1})$.

---

**Theorem 4.1.** *If the labeled data $\{(X_i, Y_i)\}_{i=1}^n$ and test data $(X_{n+1}, Y_{n+1})$ are i.i.d., then we have the finite-sample robustness guarantee*

$$\mathbb{P}\{\phi(Y_{n+1}, z_{\mathcal{U}_{\text{Cal}}}(X_{n+1})) \le r_{\mathcal{U}_{\text{Cal}}}(X_{n+1})\} \ge 1 - \alpha.$$

The finite-sample robustness relies solely on the exchangeability of data, which is identical to that in classical conformal prediction theory (Lei et al., 2018). For implementation, note that the calibrated prediction set $\mathcal{U}_{\text{Cal}}(X_{n+1})$ is obtained by traversing all possible values of $y \in \mathcal{Y}$. In practice, we can apply the discretization technique (Chen et al., 2018) to avoid exhaustive search. The complete implementation is provided in the Appendix B.4. The decision optimality of $z_{\mathcal{U}_{\text{Cal}}}$ is analyzed in the Appendix B.5, and corresponding simulation results will be provided in Section E.3.

## 5 EXPERIMENTS

In this section, we compare our proposed CRC with two baseline methods for robust decision-making: (i) CRO with conformal prediction sets (Sun et al., 2023); (ii) End-to-end (E2E) method (Chenreddy & Delage, 2024; Yeh et al., 2024) to minimize the expected risk certificate. For clarity, we refer to the application of CRC to ellipsoidal prediction sets as CRC-E, and to box prediction sets as CRC-B. The same naming convention is applied to the CRO and E2E methods for consistency. The implementation details of each baseline method are given in Appendix E.2.

We utilize the following metrics to evaluate the performance of three methods. (i) **Risk Certificate**: The average of $r_{\mathcal{U}}(X)$ across all test samples; (ii) **Decision Loss**: The average of $\phi(Y, z_{\mathcal{U}}(X))$ across all test samples; (iii) **Robustness**: The proportion of test samples where the $\phi(Y, z_{\mathcal{U}}(X))$ is less or equal to $r_{\mathcal{U}}(X)$; (iv) **Coverage**: The proportion of test samples where the true label $Y$ is covered by the prediction set $\mathcal{U}(X)$.

## 5.1 SYNTHETIC DATA ON PORTFOLIO OPTIMIZATION

In this simulation, we define the loss function as $\phi(y, z) = -y^\top z$, where $\mathcal{Y} = \mathbb{R}^2$ and $\mathcal{Z} = \{z \in [0,1]^2 : \|z\|_1 = 1\}$. The labeled data and test data are generated by:

$$Y_1 = -5X_1 - 2X_2^2 - e_1, \quad Y_2 = -3X_1^2 - X_2 - e_2,$$

where $Y = (Y_1, Y_2)$, $X = (X_1, X_2)$, and $e = (e_1, e_2)$. The covariate $X \sim N((1,1)^\top, 2.25 \cdot I_2)$, where $I_2$ is a 2-dimensional identity matrix. The noise $e \sim N(0, I_2)$ is independent of $X$. We only consider ellipsoidal prediction sets since the oracle prediction set of $Y \mid X$ is ellipsoidal under the normal noise setting. Further experimental details will be presented in Appendix E.1. All methods are evaluated over 100 trials, and the average results are reported.

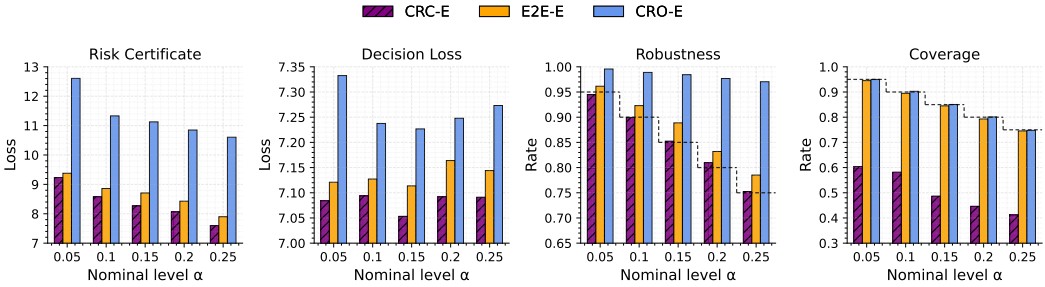

Figure 2: The results of risk certificate, decision loss, robustness, and coverage on synthetic data when varying nominal level $\alpha$ with identical sample size $n = 1500$. The horizontal gray dashed lines refer to robustness levels. The prediction sets are ellipsoids.

**Results.** We evaluate the decision performance of CRC and the baseline methods by varying the nominal level $\alpha$. As shown in Figure 2, CRC consistently outperforms the baselines in both risk certificate and decision loss. In addition, CRC also maintains the robustness level to the nominal target, while the baseline methods tend to be more conservative. With respect to coverage, CRC attains a much lower coverage rate than the robustness level, which verifies the motivation of our method. Additionally, the results for varying sample sizes are shown in Figure 3. CRC-E continues to show strong performance across all metrics, demonstrating its stable advantage. In Figure 6 of Appendix E.3, we present the density plots for risk certificate and decision loss when $\alpha = 0.15$. The overall density of CRC is shifted towards the lower loss region, further validating its superiority.

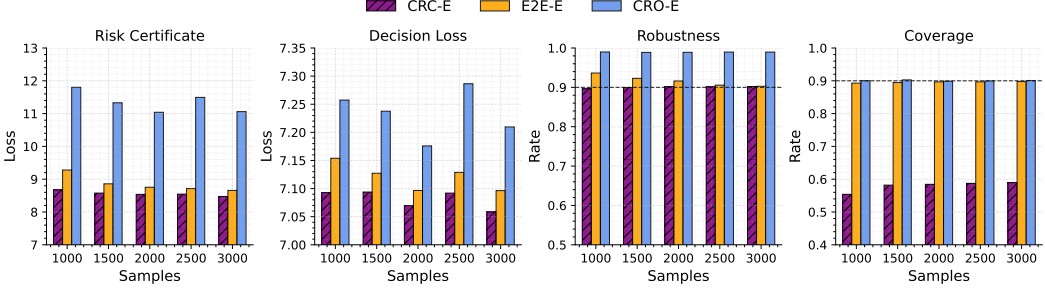

Figure 3: The results of risk certificate, decision loss, robustness, and coverage on synthetic data when varying sample size $n$ with identical nominal level $\alpha = 0.1$.

Since the RAC method proposed Kiyani et al. (2025) is applicable to discrete decision space in classification problem, we conduct the simulation on RAC method by discretizing the label space $\mathcal{Y}$ and decision space $\mathcal{Z}$. The results are provided in Appendix E.4. In addition, we also report the simulation results under the polyhedral prediction set in Appendix E.6.

## 5.2 US STOCK PROBLEM

We conduct an additional experiment on the portfolio optimization problem using a real-world dataset, following the experimental design outlined in Chenreddy et al. (2022). The dataset comprises historical US stock market data from January 1, 2012, to December 31, 2020, covering 64 stocks across eight different sectors. Daily percentage gains or losses are computed from the adjusted closing prices of consecutive trading days and used as labels. To enhance the input information for the model, we also incorporate the trading volume of individual stocks and several market benchmark indices as covariates. To evaluate the robustness of the methodology, we randomly select 15 stocks from the pool of 64 as the investable asset set in each experiment and repeat the process multiple times to mitigate the influence of random chance. We define the loss function as $\phi(y, z) = -y^\top z$, where $\mathcal{Y} = \mathbb{R}^q$ and $\mathcal{Z} = \{z \in [0, 1]^q : \|z\|_1 = 1, z \geq 0\}$.

Table 1: The results of risk certificate, decision loss, and robustness under nominal levels $\alpha = 0.1$ and $\alpha = 0.2$ on the US stock problem.

| Method | Nominal level $\alpha = 0.1$ | | | Nominal level $\alpha = 0.2$ | | |
|---|---|---|---|---|---|---|
| | Risk Certificate | Decision Loss | Robustness (%) | Risk Certificate | Decision Loss | Robustness (%) |
| CRC-B | **1.160** | **-0.055** | **90.9** | **0.731** | **-0.059** | **80.6** |
| CRO-B | 3.794 | -0.051 | 99.9 | 3.017 | -0.054 | 99.5 |
| E2E-B | 2.129 | -0.046 | 96.7 | 1.512 | -0.041 | 92.7 |
| CRC-E | **1.028** | **-0.077** | **90.8** | **0.701** | **-0.075** | **80.6** |
| CRO-E | 6.345 | -0.069 | 99.9 | 6.195 | -0.046 | 99.8 |
| E2E-E | 4.995 | -0.071 | 98.6 | 4.503 | -0.070 | 96.4 |

**Results.** As shown in Table 1, CRC outperforms the baseline methods in both risk certification and decision loss. In terms of robustness, CRC maintains a level close to the target $1 - \alpha$, demonstrating strong stability and adaptability. In contrast, E2E and CRO frequently exceed the nominal robustness target, which indicates the adoption of overly conservative strategies that lead to higher losses and risks. Overall, CRC achieves a superior balance between risk control and decision performance.

## 5.3 BATTERY STORAGE PROBLEM

In this subsection, we consider a battery storage control problem based on the frameworks of Donti et al. (2017) and Yeh et al. (2024). Given hourly electricity price forecasts $y \in \mathbb{R}^T$ and contextual covariates over a $T$-hour horizon, the controller determines the charging power $z^{\text{in}} \in \mathbb{R}^T$, discharging power $z^{\text{out}} \in \mathbb{R}^T$, and the resulting state of charge $z^{\text{state}} \in \mathbb{R}^T$, subject to the constraints for $t = 1, \ldots, T$:

$$z_0^{\text{state}} = \frac{B}{2}, \ z_t^{\text{state}} = z_{t-1}^{\text{state}} - z_t^{\text{out}} + \gamma z_t^{\text{in}},$$
$$0 \leq z_t^{\text{in}} \leq c^{\text{in}}, \ 0 \leq z_t^{\text{out}} \leq c^{\text{out}}, \ 0 \leq z_t^{\text{state}} \leq B.$$

Here $B$ denotes battery capacity, $\gamma$ denotes charging efficiency, and $c^{\text{in}}, c^{\text{out}}$ denotes per-hour power limits. The objective balances three key factors: (1) Profit from arbitrage, which involves buying and selling energy based on the prices $y \in \mathbb{R}^T$; (2) Flexibility, which is encouraged by maintaining the battery's state of charge close to half of its total capacity; (3) Battery health, which is preserved by penalizing large charging and discharging magnitudes. The resulting loss function is:

$$\phi(y, z) = \sum_{t=1}^{T} y_t \left( z_t^{\text{in}} - z_t^{\text{out}} \right) + \beta \left\| z^{\text{state}} - \frac{B}{2} \mathbf{1} \right\|_2^2 + \varepsilon \left( \|z^{\text{in}}\|_2^2 + \|z^{\text{out}}\|_2^2 \right),$$

Following Donti et al. (2017) and Yeh et al. 2024, we also set $T = 24$ hours, $B = 1$, $\gamma = 0.9$, $c^{\text{in}} = 0.5$, $c^{\text{out}} = 0.2$, $\beta = 0.1$, and $\varepsilon = 0.05$.

**Results.** Figure 4 presents a comparative analysis of the CRC method against the baselines using ellipsoidal prediction sets. For clearer visualization, negative indicator values are mapped onto the positive half-axis via a sigmoid transformation. The results demonstrate CRC's consistent superiority over both E2E and CRO across all three key metrics. As the nominal level increases, CRC effectively mitigates risk in all measures, sustaining the lowest risk and loss values at higher levels.

In addition, CRC maintains robustness values close to the nominal target, highlighting its stability and adaptability. In contrast, E2E and CRO consistently exceed the robustness target, leading to decisions characterized by excessive conservatism. The results of the box prediction set is presented in Figure 5.

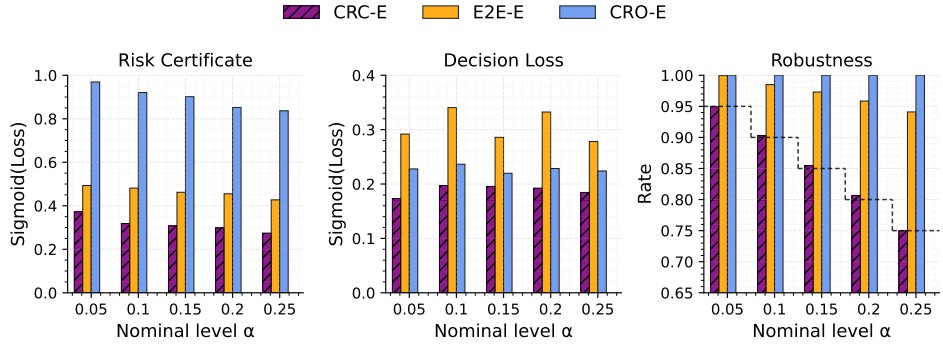

Figure 4: The results of risk certificate, decision loss, and robustness when varying nominal level $\alpha$ on battery storage problem. The prediction sets are ellipsoids.

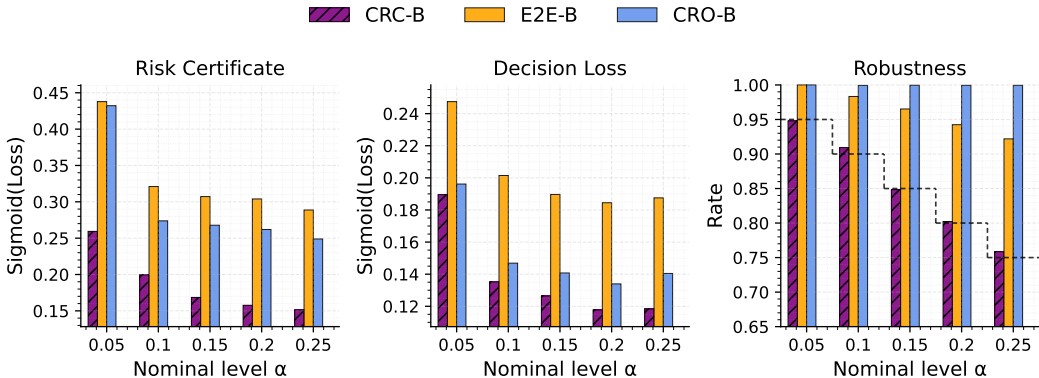

Figure 5: The risk certificate, decision loss and robustness when varying nominal level $\alpha$ on battery storage problem. The prediction sets are box.

## 6 CONCLUSION AND DISCUSSION

This paper introduces Conformal Robustness Control (CRC), a novel framework that optimizes the construction of prediction sets for robust decision-making by directly minimizing the expected risk certificate under robustness constraints. Unlike existing conditional robust optimization methods with conformal prediction sets, CRC adopts robustness constraints instead of coverage constraints, expanding the range of feasible prediction sets and enabling more efficient decisions. Theoretical guarantees for both robustness and optimality are provided, and empirical results on real-world data demonstrate significant improvements over baseline methods. Our work also identifies future research directions in data-driven robust optimization, such as developing more efficient strategies to solve the optimization problem and designing domain-specific parameterizations of prediction sets to achieve higher-quality decisions in practical applications.

### ACKNOWLEDGEMENTS

We would like to thank the anonymous reviewers and the area chair for their helpful comments and suggestions. Changliang Zou was supported by the National Key R&D Program of China

(Grant Nos. 2022YFA1003703, 2022YFA1003800), and the National Natural Science Foundation of China (Grant No. 12231011). Yajie Bao was supported by the National Natural Science Foundation of China (Grant No. 12501408), the Postdoctoral Fellowship Program of CPSF (Grant No. GZC20251996), and the fellowship from CPSF (Grant No. 2025M773046). Haojie Ren was supported by the National Key R&D Program of China (Grant No. 2024YFA1012200), the National Natural Science Foundation of China (Grant Nos. 12471262, 12522115), and Shanghai Jiao Tong University 2030 Initiative.

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

## A  USAGE STATEMENT OF LARGE LANGUAGE MODEL

We used a large language model solely for improving the fluency and readability of the manuscript. The model was not involved in research ideation, experimental design, data analysis, or result interpretation. All scientific contributions and substantive content were solely produced by the authors.

## B  MORE DISCUSSION ON RELATED WORK

### B.1  RELATIONSHIP BETWEEN THE CRC PROBLEM AND THE VAR PROBLEM

The following proposition illustrates the relationship between the VaR problem and the CRC problem (4).

**Proposition B.1.** *Let $z^Q(X) = \arg\min_{z \in \mathcal{Z}} \mathrm{VaR}_{1-\alpha}\left(\phi(Y, z) \mid X\right)$ be the unique minimizer of VaR problem. There exists a prediction set $\mathcal{U}^Q$ such that $z_{\mathcal{U}^Q} = z^Q$. Moreover, the robustness constraint is satisfied:*

$$\mathbb{P}\left[\phi\left(Y, z_{\mathcal{U}^Q}(X)\right) \leq r_{\mathcal{U}^Q}(X)\right] \geq 1 - \alpha.$$

*Furthermore, there exist cases where $z^Q = z_{\mathcal{U}^*}$, with $\mathcal{U}^*$ being the solution to optimization problem (4), and cases where $z^Q \neq z_{\mathcal{U}^*}$.*

The above conclusion indicates that, at least in some cases, decision $z_{\mathcal{U}^*}$ and decision $z^Q$ are consistent. When decision $z_{\mathcal{U}^*}$ and decision $z^Q$ are inconsistent, the expected risk certificate generated by decision $z_{\mathcal{U}^*}$ will also be lower than that of $z^Q$, indicating that $z_{\mathcal{U}^*}$ still holds practical significance.

*Proof of Proposition B.1.* To find a prediction set $\mathcal{U}^Q$ such that $z^Q$ equals $z_{\mathcal{U}^Q}$, it is sufficient to define prediction set $\mathcal{U}^Q$ in the following form:

$$\mathcal{U}^Q(x) = \left\{y \in \mathcal{Y} : \phi\left(y, z^Q(x)\right) \leq \mathrm{VaR}_{1-\alpha}\left(\phi\left(Y, z^Q(X)\right) \mid X = x\right)\right\}, \quad \forall x \in \mathcal{X}.$$

Next, we will proceed with the verification. Since the coverage constraint is a sufficient condition for the robustness constraint, we have

$$
\begin{aligned}
\mathbb{P}\left\{\phi\left(Y, z_{\mathcal{U}^Q}(X)\right) \leq r_{\mathcal{U}^Q}(X)\right\} &\geq \mathbb{P}\left\{Y \in \mathcal{U}^Q(X)\right\} \\
&= \mathbb{P}\left\{\phi\left(Y, z^Q(X)\right) \leq \mathrm{VaR}_{1-\alpha}\left(\phi\left(Y, z^Q(X)\right) \mid X\right)\right\} \\
&\geq 1 - \alpha.
\end{aligned}
$$

Thus, we verify that robustness holds. Secondly, based on the definition of quantiles, we have $\mathbb{P}\{Y \in \mathcal{U}^Q(x)\} \geq 1 - \alpha, \forall x \in \mathcal{X}$. Therefore, we can control the upper bound of $\mathrm{VaR}_{1-\alpha}\left(\phi\left(Y, z_{\mathcal{U}^Q}(X)\right) \mid X = x\right)$ in the following way:

$$
\begin{aligned}
\mathrm{VaR}_{1-\alpha}\left(\phi\left(Y, z_{\mathcal{U}^Q}(X)\right) \mid X = x\right) &\leq \max_{y \in \mathcal{U}^Q(x)} \phi\left(y, z_{\mathcal{U}^Q}(x)\right) \\
&\leq \max_{y \in \mathcal{U}^Q(x)} \phi\left(y, z^Q(x)\right) \\
&\leq \mathrm{VaR}_{1-\alpha}\left(\phi\left(Y, z^Q(X)\right) \mid X = x\right).
\end{aligned}
$$

That is, $z_{\mathcal{U}^Q}$ is also the optimal solution in the sense of minimizing the $1 - \alpha$ quantile. Therefore, $z_{\mathcal{U}^Q}$ equals $z^Q$ if $z^Q$ is the unique optimal solution.

For the scenario where decision $z^Q$ is equal to decision $z_{\mathcal{U}^*}$, consider the following example. Let $\mathcal{X} = \{0, 1\}$ and $\mathcal{Z} = \{0, 1\}$. Suppose that the density of $\phi(Y, z)$ given $X = x$ is as follows.

For $x \in \{0, 1\}, z = 0$, the density is

$$f(\phi) = \frac{1}{25}\left[3\mathbb{1}\{0 \leq \phi < 7.5\} + \mathbb{1}\{7.5 \leq \phi < 10\}\right].$$

For $x \in \{0, 1\}, z = 1$, the density is

$$f(\phi) = \frac{1}{25}\left[\mathbb{I}\{0 \leq \phi < 2.5\} + 3\mathbb{I}\{2.5 \leq \phi < 10\}\right].$$

It can be verified that when $\alpha = 0.1$, the optimal solutions of VaR and problem (4) are the same:

$$z_{\mathcal{U}^*} = z^Q = \begin{cases} 0, & \text{if } x \text{ is } 0 \\ 0, & \text{if } x \text{ is } 1 \end{cases}.$$

For the scenario where decision $z^Q$ is not equal to decision $z_{\mathcal{U}^*}$, consider the following example. Let $\mathcal{X} = \{0, 1\}$ and $\mathcal{Z} = \{0, 1\}$. Suppose that the density of $\phi(Y, z)$ given $X = x$ is as follows.

For $x \in \{0, 1\}, z = 0$, the density is

$$f(\phi) = \frac{1}{15}[\mathbb{1}\{0 \leq \phi < 1\} + 2\sum_{k=1}^{2} \mathbb{1}\{4k - 3 \leq \phi < 4k - 1\}$$

$$+ \sum_{k=1}^{2} \mathbb{1}\{4k - 1 \leq \phi < 4k + 1\} + 2\mathbb{1}\{9 \leq \phi < 10\}].$$

For $x \in \{0, 1\}, z = 1$, the density is

$$f(\phi) = \frac{1}{15}[2\mathbb{I}\{0 \leq \phi < 1\} + \sum_{k=1}^{2} \mathbb{I}\{4k - 3 \leq \phi < 4k - 1\}$$

$$+ 2\sum_{k=1}^{2} \mathbb{I}\{4k - 1 \leq \phi < 4k + 1\} + \mathbb{I}\{9 \leq \phi < 10\}].$$

It can be verified that when $\alpha = 0.4 - 4\epsilon/30$ ($\epsilon$ is sufficiently small), we have

$$z^Q = \begin{cases} 0, & \text{if } x \text{ is } 0 \\ 0, & \text{if } x \text{ is } 1 \end{cases}.$$

On the contrary, the solution of the problem (4) is different

$$z_{\mathcal{U}^*} = \begin{cases} 0, & \text{if } x \text{ is } 0 \\ 1, & \text{if } x \text{ is } 1 \end{cases}, \quad \mathcal{U}^* = \begin{cases} \{y \in \mathcal{Y} : \phi(y, 0) \leq 6.5 + 2\epsilon\}, & \text{if } x \text{ is } 0 \\ \{y \in \mathcal{Y} : \phi(y, 1) \leq 5\}, & \text{if } x \text{ is } 1 \end{cases}.$$

Note that in the example above, $\mathcal{X}$ and $\mathcal{Z}$ are discrete spaces. We can naturally extend them to the continuous spaces $[0, 1]$ while keeping the conclusions unchanged. The specific details are omitted here. $\qquad\square$

## B.2 RELATIONSHIP BETWEEN CRC AND RA-DPO, RA-CPO IN KIYANI ET AL. (2025)

For classification problems, Kiyani et al. (2025) proposed the following RA-DPO framework for the optimal decision:

$$\min_{z(\cdot),r(\cdot)} \mathbb{E}[r(X)] \quad \text{s.t.} \quad \mathbb{P}\{\phi(Y, z(X)) \leq r(X)\} \geq 1 - \alpha. \tag{8}$$

This optimization problem can be viewed as a marginal version of the VaR problem. In addition, Kiyani et al. (2025) also defined an optimal decision framework based on prediction sets, called RA-CPO, as follows:

$$\min_{\mathcal{U}():\mathcal{X}\to 2^{\mathcal{Y}}} \mathbb{E}[r_{\mathcal{U}}(X)] \quad \text{s.t.} \quad \mathbb{P}\{Y \in \mathcal{U}(X)\} \geq 1 - \alpha. \tag{9}$$

The difference between RA-CPO and CRC lies in the fact that the former employs a coverage constraint rather than a robustness constraint. We can leverage the idea from Theorem 3.2 in Kiyani et al. (2025) to prove the equivalence between the CRC problem and the RA-DPO, RA-CPO problem.

*Proof of Theorem 3.1.* Since RA-DPO and RA-CPO have been proved to be equivalent in Kiyani et al. (2025), it suffices to establish the equivalence between RA-DPO and CRC.

Let $(z^{\mathrm{RA-DPO}}(x), r^{\mathrm{RA-DPO}}(x))$ be an optimal solution to RA-DPO. Define the uncertainty set

$$\mathcal{U}^*(x) = \left\{y : \phi\left(y, z^{\mathrm{RA-DPO}}(x)\right) \leq r^{\mathrm{RA-DPO}}(x)\right\}.$$

Then,

$$\mathbb{P}\left\{\phi\left(Y, z_{\mathcal{U}^*}(X)\right) \leq r_{\mathcal{U}^*}(X)\right\}$$
$$\geq \mathbb{P}\{Y \in \mathcal{U}^*(X)\}$$
$$\geq \mathbb{P}\left\{\phi\left(Y, z^{\mathrm{RA-DPO}}(X)\right) \leq r^{\mathrm{RA-DPO}}(X)\right\} \geq 1 - \alpha.$$

Thus, $\mathcal{U}^*$ satisfies the constraint of CRC. Moreover, by definition,

$$r_{\mathcal{U}^*}(x) = \arg\min_{z \in \mathcal{Z}} \max_{y \in \mathcal{U}^*(x)} \phi(y, z)$$
$$\leq \max_{y \in \mathcal{U}^*(x)} \phi\left(y, z^{\mathrm{RA-DPO}}(x)\right) \leq r^{\mathrm{RA-DPO}}(x).$$

Hence,

$$\mathbb{E}[r_{\mathcal{U}^*}(X)] \leq \mathbb{E}[r^{\mathrm{RA-DPO}}(X)].$$

This shows that any optimal solution of RA-DPO induces a feasible solution to CRC with a risk certificate at least as good. Conversely, let $\mathcal{U}^*$ be an optimal solution to CRC. Define

$$z^{\mathrm{RA-DPO}}(x) = z_{\mathcal{U}^*}(x), \quad r^{\mathrm{RA-DPO}}(x) = r_{\mathcal{U}^*}(x).$$

This pair is feasible for RA-DPO and satisfies $\mathbb{E}[r^{\mathrm{RA-DPO}}(X)] = \mathbb{E}[r_{\mathcal{U}^*}(X)]$. Therefore, RA-DPO and CRC are equivalent, and the theorem follows. $\square$

### B.3 SUPERIORITY OF PARAMETRIZED CRC OVER PARAMETRIZED RA-CPO

The parametric formulation of RA-CPO in (9) is given by:

$$\min_{\theta \in \Theta} \mathbb{E}\left[r_\theta(X)\right] \quad \text{s.t.} \quad \mathbb{P}\left\{Y \in \mathcal{U}_\theta(X)\right\} \geq 1 - \alpha.$$

The difference between parametrized RA-CPO and parametrized CRC (5) lies in the fact that the former employs a coverage constraint rather than a robustness constraint. The relationship between the two frameworks is formalized in the following proposition.

**Proposition B.2.** *For any parameterized prediction set $\mathcal{U}_\theta(\cdot)$, it holds that*

$$\mathbb{E}[r_{\theta^{\mathrm{CRC}}}(X)] \leq \mathbb{E}[r_{\theta^{\mathrm{RA-CPO}}}(X)],$$

*where $\theta^{\mathrm{CRC}}$ and $\theta^{\mathrm{RA-CPO}}$ denote the theoretical optimal solutions of the parametrized CRC and parametrized RA-CPO problems, respectively. Moreover, there exist cases in which the inequality is strict.*

In fact, if no constraints are imposed on the prediction set, then as proven in Section B.2, the RA-CPO and CRC frameworks are equivalent. However, in regression settings, prediction sets are generally required to satisfy certain structural properties—such as convexity and boundedness—in addition to being parameterized to render the problem tractable. As a consequence, once the prediction set is parameterized, the solution derived from the CRC problem typically outperforms that obtained via RA-CPO.

*Proof.* We first show that $\mathbb{E}[r_{\theta^{\mathrm{CRC}}}(X)] \leq \mathbb{E}[r_{\theta^{\mathrm{RA-CPO}}}(X)]$. Let $\mathcal{U}_{\theta^{\mathrm{RA-CPO}}}$ be the optimal solution to the RA-CPO problem. Since it also satisfies the constraints of the CRC problem, the inequality follows directly from the definition of the CRC problem.

We now proceed to construct a case where the inequality is strict. Consider a parameterized prediction set of the form:

$$\mathcal{U}_\theta(x) = \{y \in \mathbb{R}^q : (y - \mu(x))^\top \Sigma^{-1}(x)(y - \mu(x)) \leq \theta\}, \theta \in \mathbb{R}^+,$$

where $Y \mid X \sim N(\mu(X), \Sigma(X))$. Let the loss function be $\phi(y, z) = -y^\top z$. Then the coverage probability is given by:

$$\mathbb{P}\{Y \in \mathcal{U}_\theta(X)\} = \mathbb{P}\{\chi_q^2 \leq \theta\},$$

where $\chi_q^2$ denotes a chi-squared random variable with $q$ degrees of freedom. To analyze the robustness constraint, we derive the dual of the inner maximization in the CRO problem:

$$\max_{y \in \mathcal{U}_\theta(X)} -y^\top z_\theta(X) = \sqrt{\theta}\|\Sigma^{1/2}(X)z_\theta(X)\|_2 - \mu(X)^\top z_\theta(X).$$

By the definition of the robustness level, we have:

$$\mathbb{P}\left\{\phi(Y, z_\theta(X)) \leq \max_{y \in \mathcal{U}_\theta(X)} \phi(y, z_\theta(X))\right\}$$

$$= \mathbb{P}\left\{-Y^\top z_\theta(X) \leq \sqrt{\theta}\|\Sigma^{1/2}(X)z_\theta(X)\|_2 - \mu(X)^\top z_\theta(X)\right\}$$

$$= \mathbb{P}\left\{-z_\theta(X)^\top(Y - \mu(X)) \leq \sqrt{\theta}\|\Sigma^{1/2}(X)z_\theta(X)\|_2\right\}$$

$$= \mathbb{P}\left\{\frac{-z_\theta(X)^\top(Y - \mu(X))}{\|\Sigma^{1/2}(X)z_\theta(X)\|_2} \leq \sqrt{\theta}\right\} = \mathbb{P}\{N(0,1) \leq \sqrt{\theta}\}.$$

Therefore, when $q \geq 1$, we obtain:

$$\theta^{\mathrm{CRC}} = \Phi_{1-\alpha}^2 < \chi_{q,1-\alpha}^2 = \theta^{\mathrm{RA-CPO}},$$

where $\chi_{q,1-\alpha}^2$ and $\Phi_{1-\alpha}$ denote the $(1-\alpha)$-quantiles of the $\chi_q^2$ and $N(0,1)$ distributions, respectively. In this case, it follows that:

$$\mathbb{E}[r_{\theta^{\mathrm{CRC}}}(X)] < \mathbb{E}[r_{\theta^{\mathrm{RA-CPO}}}(X)],$$

which completes the proof. $\qquad\square$

### B.4 IMPLEMENTATION OF CAL-CRC

---

**Algorithm 3** Discretization construction of Cal-CRC

---

1: **Input**: Same as Algorithm 2. Discretized space $\widetilde{\mathcal{Y}}$ with finite cardinality. Discretization mapping $\mathcal{A}(\cdot) : \mathcal{Y} \to \widetilde{\mathcal{Y}}$. Step size $\tau_0 > 0$.
2: **Discretization**: Obtain the discretized calibration set $\widetilde{\mathcal{D}}_{\mathrm{cal}} = \{(X_i, \widetilde{Y}_i)\}_{i=n_0+1}^n$ by discretization mapping $\mathcal{A}$.
3: **Calibration initialization**: $\widetilde{\mathcal{U}}_{\mathrm{Cal}}(X_{n+1}) \leftarrow \emptyset$.
4: **for** $y \in \widetilde{\mathcal{Y}}$ **do**
5: $\quad$ Define the augmented calibration set $\left\{(X_i, \widetilde{Y}_i^y)\right\}_{i=n_0+1}^{n+1}$.
6: $\quad$ $t \leftarrow 0$.
7: $\quad$ $s \leftarrow \frac{1}{n-n_0+1}\sum_{i=n_0+1}^{n+1} \mathbb{1}\left\{\phi\left(\widetilde{Y}_i^y, z_{\hat{\theta}_0,t}(X_i)\right) \leq r_{\hat{\theta}_0,t}(X_i)\right\}$.
8: $\quad$ **while** $s < 1 - \alpha$ **do**
9: $\quad\quad$ $t \leftarrow t + \tau_0$.
10: $\quad\quad$ $s \leftarrow \frac{1}{n-n_0+1}\sum_{i=n_0+1}^{n+1} \mathbb{1}\left\{\phi\left(\widetilde{Y}_i^y, z_{\hat{\theta}_0,t}(X_i)\right) \leq r_{\hat{\theta}_0,t}(X_i)\right\}$.
11: $\quad$ **end while**
12: $\quad$ $\hat{t}^y \leftarrow t$.
13: $\quad$ **if** $\phi\left(y, z_{\hat{\theta}_0,\hat{t}^y}(X_{n+1})\right) \leq r_{\hat{\theta}_0,\hat{t}^y}(X_{n+1})$ **then**
14: $\quad\quad$ $\widetilde{\mathcal{U}}_{\mathrm{Cal}}(X_{n+1}) \leftarrow \widetilde{\mathcal{U}}_{\mathrm{Cal}}(X_{n+1}) \cup \{y\}$.
15: $\quad$ **end if**
16: **end for**
17: **Anti-discretization**: $\mathcal{U}_{\mathrm{Cal}}(X_{n+1}) \leftarrow \mathcal{A}^{-1}(\widetilde{\mathcal{U}}_{\mathrm{Cal}}(X_{n+1}))$.
18: **Output**: $\mathcal{U}_{\mathrm{Cal}}(X_{n+1})$.

---

### B.5 OPTIMALITY ANALYSIS OF CAL-CRC

Under certain conditions, the discrepancy between $r_{\mathcal{U}_{\mathrm{Cal}}}(X_{n+1})$ and $r_{\mathcal{U}_{\hat{\theta}_0}}(X_{n+1})$ is expected to be negligible. For instance, if $\hat{t}^y = 0$ for any $y \in \mathcal{Y}$, then by the definition of $\mathcal{U}_{\mathrm{Cal}}$, we have $\mathcal{U}_{\hat{\theta}_0}(X_{n+1}) \subset \mathcal{U}_{\mathrm{Cal}}(X_{n+1})$. Consequently,

$$\max_{y \in \mathcal{U}_{\mathrm{Cal}}(X_{n+1})} \phi(y, z_{\mathcal{U}_{\mathrm{Cal}}}(X_{n+1})) \overset{(a)}{\geq} \max_{y \in \mathcal{U}_{\hat{\theta}_0}(X_{n+1})} \phi(y, z_{\mathcal{U}_{\mathrm{Cal}}}(X_{n+1})) \overset{(b)}{\geq} \max_{y \in \mathcal{U}_{\hat{\theta}_0}(X_{n+1})} \phi(y, z_{\mathcal{U}_{\hat{\theta}_0}}(X_{n+1})),$$

where (a) follows from the inclusion relationship between the two prediction sets, (b) holds due to the optimality of $z_{\mathcal{U}_{\hat{\theta}_0}}(X_{n+1})$ over $\mathcal{U}_{\hat{\theta}_0}(X_{n+1})$. On the other hand, from a different perspective,

$$\max_{y \in \mathcal{U}_{\mathrm{Cal}}(X_{n+1})} \phi(y, z_{\mathcal{U}_{\mathrm{Cal}}}(X_{n+1})) \overset{(c)}{\leq} \max_{y \in \mathcal{U}_{\mathrm{Cal}}(X_{n+1})} \phi(y, z_{\mathcal{U}_{\hat{\theta}_0}}(X_{n+1})) \overset{(d)}{\leq} \max_{y \in \mathcal{U}_{\hat{\theta}_0}(X_{n+1})} \phi(y, z_{\mathcal{U}_{\hat{\theta}_0}}(X_{n+1})),$$

where (c) is due to the optimality of $z_{\mathcal{U}_{\mathrm{Cal}}}(X_{n+1})$ over $\mathcal{U}_{\mathrm{Cal}}$, and (d) follows from the definition of the $\mathcal{U}_{\mathrm{Cal}}$. Combining these results yields $r_{\mathcal{U}_{\hat{\theta}_0}}(X_{n+1}) = r_{\mathcal{U}_{\mathrm{Cal}}}(X_{n+1})$. We now consider a more general setting. First, we state the generalized conditions, and then present the corresponding theoretical results.

**Condition B.1.** *Assume that for all $y \in \mathcal{Y}$, we have $|\hat{t}^y| \leq t_0$, where $t_0$ is a positive constant.*

**Condition B.2.** *Loss function $\phi$ is $L_\phi$-Lipschitz in decision $z$ for any $y \in \mathcal{Y}$. The decision $z_{\hat{\theta}_0, t}(X_{n+1})$ is $L_z$-Lipschitz in $t \leq t_0$. The risk certificate $r_{\hat{\theta}_0, t}(X_{n+1})$ is $L_r$-Lipschitz in $t \leq t_0$.*

**Theorem B.1.** *Suppose that $\hat{\theta}_0$ is obtained by running Algorithm 1 on the training dataset $\mathcal{D}_{\mathrm{train}}$. Under conditions B.1 and B.2 in the calibration process, the following result holds:*

$$r_{\mathcal{U}_{\mathrm{Cal}}}(X_{n+1}) \leq r_{\mathcal{U}_{\hat{\theta}_0}}(X_{n+1}) + t_0(L_\phi L_z + L_r).$$

*Proof.* For any $y \in \mathcal{U}_{\mathrm{Cal}}(X_{n+1})$, we have

$$\phi(y, z_{\hat{\theta}_0}(X_{n+1})) \leq \phi(y, z_{\hat{\theta}_0, \hat{t}^y}(X_{n+1})) + t_0 L_\phi L_z$$
$$\leq r_{\hat{\theta}_0, \hat{t}^y}(X_{n+1}) + t_0 L_\phi L_z$$
$$\leq r_{\mathcal{U}_{\hat{\theta}_0}}(X_{n+1}) + t_0(L_\phi L_z + L_r).$$

Therefore,

$$\max_{y \in \mathcal{U}_{\mathrm{Cal}}(X_{n+1})} \phi(y, z_{\mathrm{Cal}}(X_{n+1})) \leq \max_{y \in \mathcal{U}_{\mathrm{Cal}}(X_{n+1})} \phi(y, z_{\hat{\theta}_0}(X_{n+1}))$$
$$\leq r_{\mathcal{U}_{\hat{\theta}_0}}(X_{n+1}) + t_0(L_\phi L_z + L_r).$$

$\square$

Note that $r_{\hat{\theta}_0, t}(x)$ is monotonically increasing in $t$ for any $x \in \mathcal{X}$. Hence, if the initial model $\mathcal{U}_{\hat{\theta}_0}$ already approximately satisfies the $1 - \alpha$ robustness requirement, the calibrated threshold $\hat{t}^y$ will generally remain small for all $y \in \mathcal{Y}$. As a result, $\mathcal{U}_{\mathrm{Cal}}$ can maintain risk certificates and decision losses comparable to those of the initial model $\mathcal{U}_{\hat{\theta}_0}$. Conversely, if the initial model's robustness is significantly below $1 - \alpha$, then although $\mathcal{U}_{\mathrm{Cal}}$ still guarantee $1 - \alpha$ robustness, it may produce relatively conservative results.

## C  PROOF OF MAIN RESULTS IN SECTION 3.3

### C.1  PROOF OF THEOREM 3.2

By leveraging the finite covering property of the function class and large-sample probability inequalities, we aim to prove that the empirical estimates converge uniformly to their expected values, thereby establishing the conclusion of the theorem. First, given an $\epsilon_{2n}$-covering $\Theta_{\epsilon_{2n}}$ with smallest cardinality of the function class, and applying the Dvoretzky-Kiefer-Wolfowitz (DKW) inequality (Massart, 1990), we have

$$\mathbb{P}\left\{\sup_{t \in \mathbb{R}, \theta_0 \in \Theta_{\epsilon_{2n}}} \left|\frac{1}{n}\sum_{i=1}^{n} \mathbb{1}\{V_{\theta_0}(X_i, Y_i) \leq t\} - \mathbb{P}\{V_{\theta_0}(X, Y) \leq t\}\right| \geq \epsilon_{1n}\right\} \leq 2\mathcal{N}(\Theta, \|\cdot\|_\infty, \epsilon_{2n})e^{-2n\epsilon_{1n}^2},$$

where $\epsilon_{1n}$ is the tolerance error, whose specific value will depend on the covering number $\mathcal{N}(\Theta, \|\cdot\|_\infty, \epsilon_{2n})$ and will be specified later. According to the definition of $\epsilon_{2n}$-covering, for any given

$\theta \in \Theta$, there exists $\theta_0 \in \Theta_{\epsilon_{2n}}$ such that $\|\theta - \theta_0\| \leq \epsilon_{2n}$. Therefore, the upper bound on the deviation between the empirical estimate and the expected value can be derived as follows:

$$\left| \frac{1}{n} \sum_{i=1}^{n} \mathbb{1}\{V_\theta(X_i, Y_i) \leq 0\} - \mathbb{P}\{V_\theta(X, Y) \leq 0\} \right|$$

$$\leq \left| \frac{1}{n} \sum_{i=1}^{n} \mathbb{1}\{V_\theta(X_i, Y_i) \leq 0\} - \frac{1}{n} \sum_{i=1}^{n} \mathbb{1}\{V_{\theta_0}(X_i, Y_i) \leq 0\} \right| \qquad (10)$$

$$+ \left| \frac{1}{n} \sum_{i=1}^{n} \mathbb{1}\{V_{\theta_0}(X_i, Y_i) \leq 0\} - \mathbb{P}\{V_{\theta_0}(X, Y) \leq 0\} \right|$$

$$+ \left| \mathbb{P}\{V_{\theta_0}(X, Y) \leq 0\} - \mathbb{P}\{V_\theta(X, Y) \leq 0\} \right|. \qquad (11)$$

Leveraging the Lipschitz condition, (10) and (11) can be bounded as follows:

$$\left| \frac{1}{n} \sum_{i=1}^{n} \mathbb{1}\{V_\theta(X_i, Y_i) \leq 0\} - \frac{1}{n} \sum_{i=1}^{n} \mathbb{1}\{V_{\theta_0}(X_i, Y_i) \leq 0\} \right|$$

$$\leq \left| \frac{1}{n} \sum_{i=1}^{n} \mathbb{1}\{V_{\theta_0}(X_i, Y_i) \leq (L_\phi L_z + L_r)\|\theta - \theta_0\|\} - \frac{1}{n} \sum_{i=1}^{n} \mathbb{1}\{V_{\theta_0}(X_i, Y_i) \leq 0\} \right|$$

$$+ \left| \frac{1}{n} \sum_{i=1}^{n} \mathbb{1}\{V_{\theta_0}(X_i, Y_i) \leq -(L_\phi L_z + L_r)\|\theta - \theta_0\|\} - \frac{1}{n} \sum_{i=1}^{n} \mathbb{1}\{V_{\theta_0}(X_i, Y_i) \leq 0\} \right|$$

$$\leq 4 \sup_{t \in \mathbb{R}, \theta_0 \in \Theta_{\epsilon_{2n}}} \left| \frac{1}{n} \sum_{i=1}^{n} \mathbb{1}\{V_{\theta_0}(X_i, Y_i) \leq t\} - \mathbb{P}\{V_{\theta_0}(X, Y) \leq t\} \right|$$

$$+ \sup_{\theta_0 \in \Theta_{\epsilon_{2n}}} \mathbb{P}\{-(L_\phi L_z + L_r)\epsilon_{2n} \leq V_{\theta_0}(X, Y) \leq (L_\phi L_z + L_r)\epsilon_{2n}\},$$

and

$$\left| \mathbb{P}\{V_{\theta_0}(X, Y) \leq 0\} - \mathbb{P}\{V_\theta(X, Y) \leq 0\} \right|$$

$$\leq \left| \mathbb{P}\{V_{\theta_0}(X, Y) \leq 0\} - \mathbb{P}\{V_{\theta_0}(X, Y) \leq (L_\phi L_z + L_r)\|\theta_0 - \theta\|\} \right|$$

$$+ \left| \mathbb{P}\{V_{\theta_0}(X, Y) \leq 0\} - \mathbb{P}\{V_{\theta_0}(X, Y) \leq -(L_\phi L_z + L_r)\|\theta_0 - \theta\|\} \right|$$

$$\leq \sup_{\theta_0 \in \Theta_{\epsilon_{2n}}} \mathbb{P}\{-(L_\phi L_z + L_r)\epsilon_{2n} \leq V_{\theta_0}(X, Y) \leq (L_\phi L_z + L_r)\epsilon_{2n}\}.$$

Finally, we consolidate the above results and obtain

$$\left| \frac{1}{n} \sum_{i=1}^{n} \mathbb{1}\{V_\theta(X_i, Y_i) \leq 0\} - \mathbb{P}\{V_\theta(X, Y) \leq 0\} \right|$$

$$\leq 5 \sup_{t \in \mathbb{R}, \theta_0 \in \Theta_{\epsilon_{2n}}} \left| \frac{1}{n} \sum_{i=1}^{n} \mathbb{1}\{V_{\theta_0}(X_i, Y_i) \leq t\} - \mathbb{P}\{V_{\theta_0}(X, Y) \leq t\} \right|$$

$$+ 2 \sup_{\theta_0 \in \Theta_{\epsilon_{2n}}} \mathbb{P}\{-(L_\phi L_z + L_r)\epsilon_{2n} \leq V_{\theta_0}(X, Y) \leq (L_\phi L_z + L_r)\epsilon_{2n}\}.$$

Let $\epsilon_{1n} = \sqrt{\frac{\log(2\mathcal{N}(\Theta, \|\cdot\|_\infty, \epsilon_{2n})) + \log(1/\delta)}{2n}}$. We have

$$\sup_{\theta \in \Theta} \left| \frac{1}{n} \sum_{i=1}^{n} \mathbb{1}\{V_\theta(X_i, Y_i) \leq 0\} - \mathbb{P}\{V_\theta(X, Y) \leq 0\} \right| \leq 5\sqrt{\frac{\log(2\mathcal{N}(\Theta, \|\cdot\|_\infty, \epsilon_{2n})) + \log(1/\delta)}{2n}}$$

$$+ 4(L_\phi L_z + L_r)\rho_0 \epsilon_{2n}, \qquad (12)$$

with probability at least $1 - \delta$. Furthermore, we have, with probability at least $1 - \delta$,

$$\mathbb{P}\{\phi(Y, z_{\hat{\theta}}(X)) \leq r_{\hat{\theta}}(X) \mid \mathcal{D}_n\} \geq 1 - \alpha - 5\sqrt{\frac{\log(2\mathcal{N}(\Theta, \|\cdot\|_\infty, \epsilon_{2n})) + \log(1/\delta)}{2n}}$$

$$- 4(L_\phi L_z + L_r)\rho_0 \epsilon_{2n},$$

where $\hat{\theta}$ is the solution to the CRC problem on dataset $\mathcal{D}_n$.

## C.2 PROOF OF THEOREM 3.3

Let $\theta^*$ be the convenient notation of $\theta^*_{\Delta_n}$. The following formula gives the risk difference between the estimated model $\hat{\theta}$ and the optimal model $\theta^*$:

$$\mathbb{E}\left[r_{\hat{\theta}}(X) \mid \mathcal{D}_n\right] - \mathbb{E}\left[r_{\theta^*}(X)\right] \leq \left|\mathbb{E}[r_{\hat{\theta}}(X) \mid \mathcal{D}_n] - \frac{1}{n}\sum_{i=1}^n r_{\hat{\theta}}(X_i)\right| \tag{13}$$

$$+ \frac{1}{n}\sum_{i=1}^n r_{\hat{\theta}}(X_i) - \frac{1}{n}\sum_{i=1}^n r_{\theta^*}(X_i) \tag{14}$$

$$+ \left|\frac{1}{n}\sum_{i=1}^n r_{\theta^*}(X_i) - \mathbb{E}[r_{\theta^*}(X)]\right| \tag{15}$$

For formulas (13) and (15), we adopt a proof strategy similar to that of Theorem 3.2 to demonstrate that the empirical estimates converge uniformly to their expected value. Given $\theta \in \Theta$, let $\theta_0 \in \Theta_{\epsilon_{2n}}$ be the approximation of $\theta$ in $\epsilon_{2n}$-covering $\Theta_{\epsilon_{2n}}$. We have

$$\left|\frac{1}{n}\sum_{i=1}^n r_\theta(X_i) - \mathbb{E}[r_\theta(X)]\right| \leq \left|\frac{1}{n}\sum_{i=1}^n r_\theta(X_i) - \frac{1}{n}\sum_{i=1}^n r_{\theta_0}(X_i)\right|$$

$$+ \left|\frac{1}{n}\sum_{i=1}^n r_{\theta_0}(X_i) - \mathbb{E}[r_{\theta_0}(X)]\right|$$

$$+ |\mathbb{E}[r_{\theta_0}(X)] - \mathbb{E}[r_\theta(X)]|$$

$$\leq \sup_{\theta_0 \in \Theta_{\epsilon_{2n}}}\left|\frac{1}{n}\sum_{i=1}^n r_{\theta_0}(X_i) - \mathbb{E}[r_{\theta_0}(X)]\right|$$

$$+ 2L_r\epsilon_{2n},$$

where the last term is derived by applying the Lipschitz condition. According to Hoeffding's inequality, we have:

$$\sup_{\theta_0 \in \Theta_{\epsilon_{2n}}}\left|\frac{1}{n}\sum_{i=1}^n r_{\theta_0}(X_i) - \mathbb{E}[r_{\theta_0}(X)]\right| \leq 2B_r\sqrt{\frac{\log(2\mathcal{N}(\Theta, \|\cdot\|_\infty, \epsilon_{2n}) + \log(1/\delta)}{2n}},$$

with probability at least $1 - \delta$. Furthermore, we can derive upper bounds for formulas (13) and (15). For formula (14), we assume that event (12) in the proof of Theorem 3.2 holds. At this point, since $\hat{\theta}$ is the solution to the finite-sample CRC problem, we deduce the following result:

$$\frac{1}{n}\sum_{i=1}^n r_{\hat{\theta}}(X_i) - \frac{1}{n}\sum_{i=1}^n r_{\theta^*}(X_i) \leq 0.$$

Integrating the above conclusions, we can obtain the following result:

$$\mathbb{E}[r_{\hat{\theta}}(X) \mid \mathcal{D}_n] - \mathbb{E}[r_{\theta^*}(X)] \leq 4B_r\sqrt{\frac{\log(2\mathcal{N}(\Theta, \|\cdot\|_\infty, \epsilon_{2n}) + \log(1/\delta)}{2n}} + 4L_r\epsilon_{2n}.$$

holds with probability at least $1 - 2\delta$.

## C.3 PROOF OF THEOREM 4.1

Suppose that the calibration set is $\mathcal{D}_{\text{cal}} = \{(X_i, Y_i)\}_{i=n_0+1}^n$, the test data is $(X_{n+1}, Y_{n+1})$ and a model $\hat{\theta}_0$ has been trained from the training set $\mathcal{D}_{\text{train}} = \{(X_i, Y_i)\}_{i=1}^{n_0}$. First, we demonstrate that the prediction set $\mathcal{U}_{\text{Cal}}(\cdot)$ achieves $1 - \alpha$ coverage. Note that

$$\mathbb{P}\{Y_{n+1} \in \mathcal{U}_{\text{Cal}}(X_{n+1})\} = \mathbb{P}\left\{\phi\left(Y_{n+1}, z_{\hat{\theta}_0, \hat{t}^{Y_{n+1}}}(X_{n+1})\right) \leq r_{\hat{\theta}_0, \hat{t}^{Y_{n+1}}}(X_{n+1})\right\}.$$

Let $W = \{(X_{n_0+1}, Y_{n_0+1}), ..., (X_{n+1}, Y_{n+1})\}$ be an unordered set. Note that $\hat{t}^{Y_{n+1}}$ is measurable with respect to statistic $W$. We will complete the proof by leveraging the symmetry of the data.

$$
\mathbb{P}\left\{\phi\left(Y_{n+1}, z_{\hat{\theta}_0, \hat{t}^{Y_{n+1}}}(X_{n+1})\right) \leq r_{\hat{\theta}_0, \hat{t}^{Y_{n+1}}}(X_{n+1})\right\}
$$
$$
= \mathbb{E}\left[\mathbb{E}\left[\mathbb{1}\left\{\phi\left(Y_{n+1}, z_{\hat{\theta}_0, \hat{t}^{Y_{n+1}}}(X_{n+1})\right) \leq r_{\hat{\theta}_0, \hat{t}^{Y_{n+1}}}(X_{n+1})\right\} \mid W\right]\right]
$$
$$
= \mathbb{E}\left[\frac{1}{n - n_0 + 1}\sum_{i=n_0+1}^{n+1}\mathbb{1}\left\{\phi\left(Y_i, z_{\hat{\theta}_0, \hat{t}^{Y_{n+1}}}(X_i)\right) \leq r_{\hat{\theta}_0, \hat{t}^{Y_{n+1}}}(X_i)\right\}\right]
$$
$$
\geq 1 - \alpha.
$$

The first equality stems from the law of total expectation. The second equality arises from the symmetry of the data, a technique frequently employed in proofs within conformal prediction methods(Vovk et al., 2005; Liang et al., 2024). The final inequality is derived from the definition of threshold $\hat{t}^{Y_{n+1}}$, as referenced in Algorithm 2. Finally, since the coverage constraint is a sufficient condition for the robustness constraint, we can obtain the robustness guarantee, i.e.,

$$
\mathbb{P}\left\{\phi\left(Y_{n+1}, z_{\mathcal{U}_{\mathrm{Cal}}}(X_{n+1})\right) \leq r_{\mathcal{U}_{\mathrm{Cal}}}(X_{n+1})\right\} \geq \mathbb{P}\{Y_{n+1} \in \mathcal{U}_{\mathrm{Cal}}(X_{n+1})\} \geq 1 - \alpha.
$$

# D ADDITIONAL THEORETICAL RESULTS

## D.1 THEORETICAL RESULTS FOR VC/RADEMACHER CLASS

In this section, we present theoretical results on robustness and optimality when the function class has a finite VC dimension. Additionally, we discuss a decision-making method based on partitioning the covariate domain (Chenreddy et al., 2022). Under this approach, the corresponding function class possesses a finite VC dimension, thereby exhibiting relevant convergence properties. Let $\mathrm{VC}(\mathcal{C}) := \mathrm{VC}(\{(x,y) \to \mathbb{1}\{\phi(y, z_\theta(x)) \leq r_\theta(x)\} : \theta \in \Theta\})$ denote the VC dimension of the robustness-induced classifier class.

**Theorem D.1** (VC class robustness). *Suppose $VC(\mathcal{C}) = H < \infty$. Then there exists an absolute constant $C > 0$ such that, with probability at least $1 - \delta$,*

$$
\mathbb{P}\left\{\phi\left(Y, z_{\hat{\theta}}(X)\right) \leq r_{\hat{\theta}}(X) \mid \mathcal{D}_n\right\} \geq 1 - \alpha - C\sqrt{\frac{H}{n}} - \sqrt{\frac{\log(2/\delta)}{2n}}.
$$

*Proof.* By McDiarmid's Inequality, with probability at least $1 - \delta$,

$$
\sup_{\theta \in \Theta}\left|\frac{1}{n}\sum_{i=1}^{n}\mathbb{1}\{\phi(Y_i, z_\theta(X_i)) \leq r_\theta(X_i)\} - \mathbb{P}\{\phi(Y, z_\theta(X)) \leq r_\theta(X)\}\right|
$$
$$
\leq \mathbb{E}\left[\sup_{\theta \in \Theta}\left|\frac{1}{n}\sum_{i=1}^{n}\mathbb{1}\{\phi(Y_i, z_\theta(X_i)) \leq r_\theta(X_i)\} - \mathbb{P}\{\phi(Y, z_\theta(X)) \leq r_\theta(X)\}\right|\right] \quad (16)
$$
$$
+ \sqrt{\frac{\log(2/\delta)}{2n}}.
$$

The expectation in (16) can be bounded using the standard VC-class Rademacher bounds (Vershynin, 2018, Theorem 8.3.23): there exists a constant $C$ such that

$$
\mathbb{E}\left[\sup_{\theta \in \Theta}\left|\frac{1}{n}\sum_{i=1}^{n}\mathbb{1}\{\phi(Y_i, z_\theta(X_i)) \leq r_\theta(X_i)\} - \mathbb{P}\{\phi(Y, z_\theta(X)) \leq r_\theta(X)\}\right|\right] \leq C\sqrt{\frac{H}{n}}.
$$

Combining these results yields that, with probability at least $1 - \delta$:

$$
\mathbb{P}\left\{\phi\left(Y, z_{\hat{\theta}}(X)\right) \leq r_{\hat{\theta}}(X) \mid \mathcal{D}_n\right\} \geq 1 - \alpha - C\sqrt{\frac{H}{n}} - \sqrt{\frac{\log(2/\delta)}{2n}}.
$$

$\square$

**Theorem D.2** (Rademacher risk). *Assume additionally that $|r_\theta(x)| \leq M$ for all $\theta \in \Theta$ and $x \in \mathcal{X}$. Let $\theta^*_{\Delta_n}$ be the optimal solution of problem* (5) *at robustness level $1 - \alpha + \Delta_n$ where $\Delta_n = C\sqrt{\frac{H}{n}} + \sqrt{\frac{\log(2/\delta)}{2n}}$. Then, with probability at least $1 - 2\delta$,*

$$\mathbb{E}\left[r_{\hat{\theta}}(X) - r_{\theta^*}(X) \mid \mathcal{D}_n\right] \leq 4\mathcal{R}_n(\{r_\theta(\cdot) : \theta \in \Theta\}) + 2M\sqrt{\frac{\log(4/\delta)}{2n}},$$

*where $\mathcal{R}_n(\{r_\theta(\cdot) : \theta \in \Theta\})$ denotes the Rademacher complexity for function class $\{r_\theta(\cdot) : \theta \in \Theta\}$.*

*Proof.* Let $\theta^* = \theta^*_{\Delta_n}$. We bound the risk difference between the estimated model $\hat{\theta}$ and the optimal model $\theta^*$ as follows:

$$\mathbb{E}\left[r_{\hat{\theta}}(X) \mid \mathcal{D}_n\right] - \mathbb{E}\left[r_{\theta^*}(X)\right] \leq \left|\mathbb{E}[r_{\hat{\theta}}(X) \mid \mathcal{D}_n] - \frac{1}{n}\sum_{i=1}^n r_{\hat{\theta}}(X_i)\right|$$

$$+ \frac{1}{n}\sum_{i=1}^n r_{\hat{\theta}}(X_i) - \frac{1}{n}\sum_{i=1}^n r_{\theta^*}(X_i)$$

$$+ \left|\frac{1}{n}\sum_{i=1}^n r_{\theta^*}(X_i) - \mathbb{E}[r_{\theta^*}(X)]\right|$$

$$\leq 2\sup_{\theta \in \Theta}\left|\frac{1}{n}\sum_{i=1}^n r_\theta(X_i) - \mathbb{E}\left[r_\theta(X)\right]\right| \quad (17)$$

$$+ \frac{1}{n}\sum_{i=1}^n r_{\hat{\theta}}(X_i) - \frac{1}{n}\sum_{i=1}^n r_{\theta^*}(X_i). \quad (18)$$

For the term (17), by McDiarmid's inequality, with probability $1 - \delta$:

$$\sup_{\theta \in \Theta}\left|\frac{1}{n}\sum_{i=1}^n r_\theta(X_i) - \mathbb{E}\left[r_\theta(X)\right]\right| \leq \mathbb{E}\left[\sup_{\theta \in \Theta}\left|\frac{1}{n}\sum_{i=1}^n r_\theta(X_i) - \mathbb{E}\left[r_\theta(X)\right]\right|\right] + 2M\sqrt{\frac{\log(2/\delta)}{2n}}. \quad (19)$$

The expectation in (19) is bounded via Rademacher complexity:

$$\mathbb{E}\left[\sup_{\theta \in \Theta}\left|\frac{1}{n}\sum_{i=1}^n r_\theta(X_i) - \mathbb{E}\left[r_\theta(X)\right]\right|\right] \leq 2\mathbb{E}\left[\sup_{\theta \in \Theta}\left|\frac{1}{n}\sum_{i=1}^n \epsilon_i r_\theta(X_i)\right|\right] = 2\mathcal{R}_n(\{r_\theta(\cdot) : \Theta\}).$$

For the term (18), whenever

$$\left|\frac{1}{n}\sum_{i=1}^n \mathbb{1}\{\phi\left(Y_i, z_{\theta^*}(X_i)\right) \leq r_{\theta^*}(X_i)\} - \mathbb{P}\left\{\phi\left(Y, z_{\theta^*}(X)\right) \leq r_{\theta^*}(X)\right\}\right| \leq C\sqrt{\frac{H}{n}} + \sqrt{\frac{\log(2/\delta)}{2n}}$$

the definition of problem (6) implies

$$\frac{1}{n}\sum_{i=1}^n r_{\hat{\theta}}(X_i) - \frac{1}{n}\sum_{i=1}^n r_{\theta^*}(X_i) \leq 0.$$

By Theorem D.1, this event holds with probability at least $1 - \delta$. A union bound gives that, with probability at least $1 - 2\delta$:

$$\mathbb{E}\left[r_{\hat{\theta}}(X) \mid \mathcal{D}_n\right] - \mathbb{E}\left[r_{\theta^*}(X)\right] \leq 4\mathcal{R}_n(\{r_\theta(\cdot) : \theta \in \Theta\}) + 2M\sqrt{\frac{\log(2/\delta)}{2n}}.$$

$\square$

**Remark D.1.** *In Chenreddy et al. (2022), the authors introduce a decision-making framework that leverages data-driven learning of underlying structures to categorize individuals into $K$ classes based on their covariates. For each class, a prediction set is constructed, which in turn induces*

*specific decisions and risk certificates. Denote the trained classifier by $\mathcal{A} : \mathcal{X} \to [K]$ and the model parameters by $\theta$. The decisions and risk certificates take the following forms:*

$$z_\theta(x) = \sum_{k=1}^{K} z_\theta^k \mathbb{1}\{\mathcal{A}(x) = k\} \qquad r_\theta(x) = \sum_{k=1}^{K} r_\theta^k \mathbb{1}\{\mathcal{A}(x) = k\},$$

*where $z_k \in \mathcal{Z}, r_k \in \mathbb{R}$ for each $k \in [K]$. Considering a portfolio optimization problem with loss function $\phi(y, z) = -y^\top z$ and $\mathcal{Y} = \mathbb{R}^q$, the set $\{(x, y) \to \mathbb{1}\{\phi(y, z_\theta(x)) \le r_\theta(x)\} : \theta \in \Theta\}$ becomes a subset of the following family:*

$$\left\{ (x, y) \to \mathbb{1}\left\{ \sum_{k=1}^{K} \left( a_k^\top y - r_k \right) \mathbb{1}\{\mathcal{A}(x) = k\} \le 0 \right\} : a_k \in \mathbb{R}^q, r_k \in \mathbb{R} \text{ for all } k \in [K] \right\}.$$

*This family corresponds to a finite-dimensional linear space of functions and therefore has VC dimension at most $(q + 1)K$. Applying Theorem D.1, we obtain the following convergence guarantee: with probability at least $1 - \delta$,*

$$\mathbb{P}\left\{ \phi\left(Y, z_{\hat{\theta}}(X)\right) \le r_{\hat{\theta}}(X) \mid \mathcal{D}_n \right\} \ge 1 - \alpha - C\sqrt{\frac{(q + 1)K}{n}} - \sqrt{\frac{\log(2/\delta)}{2n}},$$

*Similarly, the function class $\{r_\theta(x) : \theta \in \Theta\}$ is uniformly bounded and has VC dimension at most $K + 1$. Hence, its Rademacher complexity satisfies $\mathcal{R}_n(\{r_\theta : \theta \in \Theta\}) \le C'\sqrt{\frac{K+1}{n}}$ for some constant $C'$. This leads to the following bound on the excess risk: with probability at least $1 - 2\delta$,*

$$\mathbb{E}\left[ r_{\hat{\theta}}(X) - r_{\theta^*}(X) \mid \mathcal{D}_n \right] \le 4C'\sqrt{\frac{K + 1}{n}} + 2M\sqrt{\frac{\log(2/\delta)}{2n}}.$$

*It is worth noting that, under the finite VC dimension condition, the resulting convergence rate achieves the order $O(\sqrt{1/n})$.*

## D.2 THEORETICAL RESULTS UNDER SMOOTH CONSTRAINT

In this section, we we analyze the theoretical properties of the optimal solution to the following smoothed optimization problem:

$$\hat{\theta} = \arg\min_{\theta \in \Theta} \frac{1}{n} \sum_{i=1}^{n} r_\theta(X_i) \quad \text{s.t.} \quad \frac{1}{n} \sum_{i=1}^{n} \tilde{\mathbb{1}}\left\{ \phi\left(Y_i, z_\theta(X_i)\right) \le r_\theta(X_i) \right\} \ge 1 - \alpha. \tag{20}$$

Here, $\tilde{\mathbb{1}}\{a \le b\} = \frac{1}{2}(1 + \text{erf}(\frac{b-a}{\sqrt{2}\sigma}))$, where $\text{erf}(x) = \frac{2}{\sqrt{\pi}} \int_0^x e^{-t^2} dt$ is the Gaussian error function and $\sigma$ controls the smoothness of the surrogate. This formulation provides a smoothed approximation of (6) and serves as the direct optimization target in Algorithm 1. The next two theorems provide a non-asymptotic guarantees of the robustness and expected risk certificate value of the resulting solution.

**Theorem D.3** (**Robustness**). *Let $\Theta_\epsilon$ denote an $\epsilon$-covering of the $\Theta$ with coverage number $\mathcal{N}(\Theta, \|\cdot\|_\infty, \epsilon)$, and let $\hat{\theta}$ be the solution of problem* (20). *Under Conditions 3.1-3.2, for any independent data $(X, Y) \sim P$ and conditioning on the labeled data $\mathcal{D}_n$, we have*

$$\mathbb{P}\left\{ \phi\left(Y, z_{\hat{\theta}}(X)\right) \le r_{\hat{\theta}}(X) \mid \mathcal{D}_n \right\} \ge 1 - \alpha - \sqrt{\frac{\log(2\mathcal{N}(\Theta, \|\cdot\|_\infty, \epsilon) + \log(1/\delta)}{2n}}$$
$$- \frac{2(L_z L_\phi + L_r)\epsilon}{\sqrt{2\pi}\sigma} - \sqrt{\frac{\pi}{2}}\sigma\rho_0,$$

*with probability at least $1 - \delta$.*

*Proof.* For any $\theta \in \Theta$, let $\theta_0 \in \Theta_\epsilon$ such that $\|\theta - \theta_0\| < \epsilon$. We decompose the deviation as follows:

$$\left| \frac{1}{n} \sum_{i=1}^{n} \tilde{\mathbb{1}} \left\{ \phi\left(Y_i, z_\theta(X_i)\right) \leq r_\theta(X_i) \right\} - \mathbb{E}\left[ \tilde{\mathbb{1}} \left\{ \phi\left(Y, z_\theta(X)\right) \leq r_\theta(X) \right\} \right] \right|$$

$$\leq \left| \frac{1}{n} \sum_{i=1}^{n} \tilde{\mathbb{1}} \left\{ \phi\left(Y_i, z_\theta(X_i)\right) \leq r_\theta(X_i) \right\} - \frac{1}{n} \sum_{i=1}^{n} \tilde{\mathbb{1}} \left\{ \phi\left(Y_i, z_{\theta_0}(X_i)\right) \leq r_{\theta_0}(X_i) \right\} \right| \quad (21)$$

$$+ \left| \frac{1}{n} \sum_{i=1}^{n} \tilde{\mathbb{1}} \left\{ \phi\left(Y_i, z_{\theta_0}(X_i)\right) \leq r_{\theta_0}(X_i) \right\} - \mathbb{E}\left[ \tilde{\mathbb{1}} \left\{ \phi\left(Y, z_{\theta_0}(X)\right) \leq r_{\theta_0}(X) \right\} \right] \right| \quad (22)$$

$$+ \left| \mathbb{E}\left[ \tilde{\mathbb{1}} \left\{ \phi\left(Y, z_{\theta_0}(X)\right) \leq r_{\theta_0}(X) \right\} \right] - \mathbb{E}\left[ \tilde{\mathbb{1}} \left\{ \phi\left(Y, z_\theta(X)\right) \leq r_\theta(X) \right\} \right] \right| \quad (23)$$

We can apply the Lipschitz condition to bound term (21):

$$\left| \frac{1}{n} \sum_{i=1}^{n} \tilde{\mathbb{1}} \left\{ \phi\left(Y_i, z_\theta(X_i)\right) \leq r_\theta(X_i) \right\} - \frac{1}{n} \sum_{i=1}^{n} \tilde{\mathbb{1}} \left\{ \phi\left(Y_i, z_{\theta_0}(X_i)\right) \leq r_{\theta_0}(X_i) \right\} \right|$$

$$\leq \frac{1}{n} \sum_{i=1}^{n} \left| \tilde{\mathbb{1}} \left\{ \phi\left(Y_i, z_\theta(X_i)\right) \leq r_\theta(X_i) \right\} - \tilde{\mathbb{1}} \left\{ \phi\left(Y_i, z_{\theta_0}(X_i)\right) \leq r_{\theta_0}(X_i) \right\} \right|$$

$$\overset{(a)}{\leq} \frac{1}{n\sigma\sqrt{2\pi}} \sum_{i=1}^{n} \left| \phi\left(Y_i, z_\theta(X_i)\right) - \phi\left(Y_i, z_{\theta_0}(X_i)\right) \right| + \left| r_\theta(X_i) - r_{\theta_0}(X_i) \right|$$

$$\overset{(b)}{\leq} \frac{(L_z L_\phi + L_r)\epsilon}{\sqrt{2\pi}\sigma},$$

where $(a)$ is due to the fact that function $\mathbb{1}\{\cdot, \cdot\}$ is $\frac{1}{\sigma\sqrt{2\pi}}$-Lipschitz continuous with respect to its both components, and $(b)$ is derived from condition 3.1. For the term (23), we can apply the same method to derive its upper bound:

$$\left| \mathbb{E}\left[ \tilde{\mathbb{1}} \left\{ \phi\left(Y, z_{\theta_0}(X)\right) \leq r_{\theta_0}(X) \right\} \right] - \mathbb{E}\left[ \tilde{\mathbb{1}} \left\{ \phi\left(Y, z_\theta(X)\right) \leq r_\theta(X) \right\} \right] \right|$$

$$\leq \mathbb{E}\left[ \left| \tilde{\mathbb{1}} \left\{ \phi\left(Y, z_{\theta_0}(X)\right) \leq r_{\theta_0}(X) \right\} - \tilde{\mathbb{1}} \left\{ \phi\left(Y, z_\theta(X)\right) \leq r_\theta(X) \right\} \right| \right]$$

$$\leq \frac{1}{\sqrt{2\pi}\sigma} \mathbb{E}\left[ \left| \phi\left(Y, z_\theta(X)\right) - \phi\left(Y, z_{\theta_0}(X)\right) \right| + \left| r_\theta(X) - r_{\theta_0}(X) \right| \right]$$

$$\leq \frac{(L_z L_\phi + L_r)\epsilon}{\sqrt{2\pi}\sigma}.$$

For term (22), by Hoeffding's inequality and a union bound over $\theta_0 \in \Theta_\epsilon$, with probability at least $1 - \delta$,

$$\sup_{\theta_0 \in \Theta_\epsilon} \left| \frac{1}{n} \sum_{i=1}^{n} \tilde{\mathbb{1}} \left\{ \phi\left(Y_i, z_{\theta_0}(X_i)\right) \leq r_{\theta_0}(X_i) \right\} - \mathbb{E}\left[ \tilde{\mathbb{1}} \left\{ \phi\left(Y, z_{\theta_0}(X)\right) \leq r_{\theta_0}(X) \right\} \right] \right|$$

$$\leq \sqrt{\frac{\log(2\mathcal{N}(\Theta, \|\cdot\|_\infty, \epsilon)) + \log(1/\delta)}{2n}}.$$

Combining these bounds yields:

$$\mathbb{E}\left[ \tilde{\mathbb{1}} \left\{ \phi\left(Y, z_{\hat{\theta}}(X)\right) \leq r_{\hat{\theta}}(X) \right\} \mid \mathcal{D}_n \right] \geq 1 - \alpha - \sqrt{\frac{\log(2\mathcal{N}(\Theta, \|\cdot\|_\infty, \epsilon)) + \log(1/\delta)}{2n}} - \frac{2(L_z L_\phi + L_r)\epsilon}{\sqrt{2\pi}\sigma}$$

with probability at least $1 - \delta$. Finally, we quantify the discrepancy between the robustness $\mathbb{E}\left[\mathbb{1}\{\phi\left(Y, z_\theta(X)\right) \leq r_\theta(X)\}\right]$ and its smoothed version $\mathbb{E}\left[\tilde{\mathbb{1}}\{\phi\left(Y, z_\theta(X)\right) \leq r_\theta(X)\}\right]$. Let $f_\theta(\cdot)$

denote the density of $V_\theta(X, Y)$. Then:

$$\mathbb{E}\left[\left|\tilde{\mathbb{1}}\{\phi(Y, z_\theta(X)) \leq r_\theta(X)\} - \mathbb{1}\{\phi(Y, z_\theta(X)) \leq r_\theta(X)\}\right|\right]$$

$$= \int_{-\infty}^0 \left(1 - \frac{1}{2}\left(1 + \mathrm{erf}(\frac{-t}{\sqrt{2}\sigma})\right)\right) f_\theta(t)dt + \int_0^{+\infty} \frac{1}{2}\left(1 + \mathrm{erf}(\frac{-t}{\sqrt{2}\sigma})\right) f_\theta(t)dt$$

$$\overset{(a)}{\leq} \frac{\rho_0}{2}\int_{-\infty}^0 1 - \mathrm{erf}(\frac{-t}{\sqrt{2}\sigma})dt + \frac{\rho_0}{2}\int_0^{+\infty} 1 + \mathrm{erf}(\frac{-t}{\sqrt{2}\sigma})dt$$

$$\overset{(b)}{\leq} \sqrt{\frac{\pi}{2}}\sigma\rho_0,$$

where $(a)$ follows from the bounded density assumption 3.2, and $(b)$ is derived via standard Gaussian integral identities. Incorporating this bound into the previous result, we conclude that with probability at least $1 - \delta$,

$$\mathbb{P}\left\{\phi\left(Y, z_{\hat\theta}(X)\right) \leq r_{\hat\theta}(X) \mid \mathcal{D}_n\right\} \geq 1 - \alpha - \sqrt{\frac{\log(2\mathcal{N}(\Theta, \|\cdot\|_\infty, \epsilon)) + \log(1/\delta)}{2n}}$$

$$- \frac{2(L_z L_\phi + L_r)\epsilon}{\sqrt{2\pi}\sigma} - \sqrt{\frac{\pi}{2}}\sigma\rho_0.$$

$\square$

The key difference from the non-smoothed case is the presence of the term $\sqrt{\frac{\pi}{2}}\sigma\rho_0$, which quantifies the bias introduced by the smoothing. Below, we directly present the relevant optimality theorem, as its proof and conclusions are almost identical to the non-smoothed case.

**Theorem D.4 (Optimality).** *Let $\theta^*_{\Delta_n}$ be the optimal solution of problem* (5) *at the robustness level* $1 - \alpha + \Delta_n$ *where* $\Delta_n = \sqrt{\frac{\log(2\mathcal{N}(\Theta, \|\cdot\|_\infty, \epsilon)) + \log(1/\delta)}{2n}} + \frac{2(L_z L_\phi + L_r)\epsilon}{\sqrt{2\pi}\sigma} + \sqrt{\frac{\pi}{2}}\sigma\rho_0$. *Under the same conditions of Theorem 3.3, conditioning on $\mathcal{D}_n$, we have*

$$\mathbb{E}\left[r_{\hat\theta}(X) - r_{\theta^*_{\Delta_n}}(X) \mid \mathcal{D}_n\right] \leq 4B_r\sqrt{\frac{\log(2\mathcal{N}(\Theta, \|\cdot\|_\infty, \epsilon)) + \log(1/\delta)}{2n}} + \frac{4L_r}{n},$$

*with probability at least $1 - 2\delta$.*

# E EXPERIMENTAL DETAILS AND ADDITIONAL RESULTS

## E.1 EXPERIMENTAL DETAILS OF CRC

To accelerate the alternating optimization of CRC and promote stable convergence, we partition the labeled data into two mutually exclusive parts: the first part is used for pretraining CRC, with the resulting parameters serving as initialization for subsequent alternating optimization; the second part is exclusively dedicated to the alternating optimization phase.

The baseline method employs the same partitioning strategy: the first portion trains the prediction model, while the second portion is used for calibration or solving downstream optimization tasks. To ensure comparability, all methods uniformly employ the same scoring function and optimization objective in experiments.

**Pre-training** Pre-training of the CRC can be approached in two ways depending on the shape of the prediction set: For ellipsoidal prediction sets, the neural network outputs the parameters of a multivariate Gaussian, namely the mean vector $\hat\mu(\cdot)$ and the covariance matrix $\hat\Sigma(\cdot)$. We parameterize $\hat\Sigma(\cdot)$ via a Cholesky factorization, $\hat\Sigma(\cdot) = L(\cdot)L(\cdot)^\top$, where $L(\cdot)$ is lower triangular. To guarantee positive definiteness, we add a small diagonal jitter to the predicted covariance, i.e., $\Sigma' = \Sigma + \varepsilon I$ which raises the eigenvalue floor and ensures numerical stability of the Cholesky factorization. Additionally, our training objective is to maximize the Gaussian log-likelihood, equivalently to minimize the negative log-likelihood:

$$\mathcal{L}_\theta = \frac{1}{(2\pi)^{\frac{d}{2}}|\Sigma|^{\frac{1}{2}}}\exp\left(-\tfrac{1}{2}(y - \mu)^\top\Sigma^{-1}(y - \mu)\right).$$

For box prediction sets, we use quantile regression to directly estimate quantiles. Concretely, we train a neural network $f_\theta(x)$ to output the $\alpha$-level quantile for input $x$. The $1 - \alpha$ confidence interval is constructed as $\left[ f_\theta^{\alpha/2}(x), \ f_\theta^{1-\alpha/2}(x) \right]$. Benefiting from quantile regression, our training objective is to minimize pinball loss. Given a quantile level $\alpha \in (0, 1)$ and prediction $\hat{y} = f_\theta(x)$, the loss for target $y$ is

$$\mathcal{L}_\alpha(y, \hat{y}) = \begin{cases} \alpha \left( y - \hat{y} \right), & \text{if } y > \hat{y}, \\ (1 - \alpha) \left( \hat{y} - y \right), & \text{if } y \leq \hat{y}. \end{cases}$$

**Optimization**  For CRC optimization, we use the `cvxpylayers` (Agrawal et al., 2019) Python package to implement the implicit function differentiation. The optimization is performed using the Adam optimizer, and we select the optimal combination of learning rates ($1e-2$, $1e-3$, $1e-4$) and L2 weight decay values ($0$, $1e-2$, $1e-3$) to minimize the optimization loss. Moreover, to mitigate overfitting and ineffective training, 20% of the data used for optimization is held out as a validation set. Early stopping is triggered when the loss on the validation set fails to decrease for 10 consecutive iterations or when the predefined maximum number of iterations is reached.

**Smoothing parameters sensitivity**  For CRC method, we approximate the indicator with a smooth surrogate $\mathbb{1}\{a \leq b\} = \frac{1}{2}(1 + \text{erf}(\frac{b-a}{\sqrt{2}\sigma}))$. We compared the sensitivity of different smoothing parameters $\sigma$ on CRC. The experimental results are summarized in Table 2.

Table 2: The results of different smoothing parameters sensitivity of CRC at the nominal level $\alpha = 0.1$, where the sample size is $n = 1500$. The prediction sets are ellipsoids.

| Method | Smoothing parameter $\sigma$ | Risk Certificate | Decision Loss | Robustness (%) | Coverage (%) |
|--------|------------------------------|------------------|---------------|----------------|--------------|
|        | 0.01 | $8.678 \pm 0.299$ | $7.072 \pm 0.220$ | $89.8 \pm 0.7$ | $60.8 \pm 5.6$ |
|        | 0.05 | $8.633 \pm 0.295$ | $7.070 \pm 0.219$ | $89.9 \pm 0.6$ | $59.6 \pm 5.5$ |
| CRC-E  | 0.10 | $8.641 \pm 0.306$ | $7.071 \pm 0.221$ | $90.3 \pm 0.5$ | $59.4 \pm 5.7$ |
|        | 0.15 | $8.643 \pm 0.315$ | $7.070 \pm 0.221$ | $90.5 \pm 0.6$ | $59.6 \pm 5.7$ |
|        | 0.20 | $8.649 \pm 0.308$ | $7.070 \pm 0.220$ | $90.2 \pm 0.5$ | $59.6 \pm 5.7$ |

**Lagrange multiplier update schedule sensitivity**  For dual variable $\lambda$, we investigated the results of CRC on Lagrange multiplier update schedule sensitivity which refers to the number of model parameter optimization steps performed before each update of $\lambda$. The experimental results will be shown in Table 3.

Table 3: The results of lagrange multiplier update schedule of CRC at the nominal level $\alpha = 0.1$, where the sample size is $n = 1500$. The prediction sets are ellipsoids.

| Method | $\lambda$ update schedule | Risk Certificate | Decision Loss | Robustness (%) | Coverage (%) |
|--------|---------------------------|------------------|---------------|----------------|--------------|
|        | 1 | $8.641 \pm 0.334$ | $7.109 \pm 0.251$ | $89.9 \pm 0.5$ | $58.7 \pm 5.8$ |
| CRC-E  | 2 | $8.528 \pm 0.302$ | $7.106 \pm 0.251$ | $90.4 \pm 0.6$ | $56.4 \pm 5.5$ |
|        | 4 | $8.478 \pm 0.278$ | $7.105 \pm 0.251$ | $90.1 \pm 0.5$ | $55.4 \pm 4.8$ |
|        | 8 | $8.452 \pm 0.282$ | $7.105 \pm 0.251$ | $89.7 \pm 0.6$ | $55.2 \pm 4.9$ |

### E.2  BASELINE METHODS

**CRO**  The CRO method is our implementation of the Predict-then-Calibrate framework proposed by Sun et al. (2023). Specifically, we first train a predictive model to parameterize the uncertainty set (e.g., outputting the mean and covariance of ellipsoidal prediction sets). Subsequently, we construct a prediction set on the calibration set that satisfies the target coverage requirement. Finally, the prediction set is directly embedded into a downstream robust optimization problem to solve for decisions and minimize task loss. Thus, this method reduces task loss while enhancing solution stability, all while ensuring coverage.

**E2E** E2E is an end-to-end robust optimization method proposed by Chenreddy & Delage (2024) and Yeh et al. (2024). Unlike CRO, E2E aims to bridge uncertainty calibration with downstream task objectives by minimizing target loss through global optimization. Specifically, E2E first trains a prediction model capable of outputting parameters of uncertainty sets. It then computes non-conformity scores on the calibration set, determines the threshold $q$ that satisfies the nominal coverage $1-\alpha$, and constructs the uncertainty set accordingly. Finally, under this uncertainty set, the robust optimization problem is solved to obtain the current task loss. The gradients of the task loss with respect to model parameters are backpropagated through the differentiable optimization layer to the prediction model, enabling collaborative updates of model parameters and task objectives. Consequently, the model achieves better alignment with real-world decisions while ensuring coverage and reducing task loss. For a fair comparison, we set the loss function in E2E method as the expected risk certificate.

### E.3 Density Plot of Simulation in Section 5.1

The density plots of the risk certificate of three methods are given in Figure 6. Compared with other baseline methods, CRC has achieved the best performance.

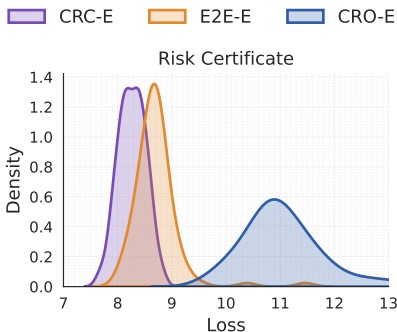

Figure 6: The densities of risk certificate on synthetic data when $\alpha = 0.15$ and $n = 1500$.

### E.4 Comparison Results of RAC and CRC

Based on the RA-CPO/RA-DPO framework, Kiyani et al. (2025) proposed the Risk-Averse Calibration (RAC) method to solve decision-making problems in classification settings. However, this method strictly relies on the finiteness of the label space and the decision space, i.e., $|\mathcal{Y}| < \infty, |\mathcal{Z}| < \infty$. Consequently, the RAC method is more suitable for classification problems and is not applicable to regression problems since constructing the prediction set in Kiyani et al. (2025) requires solving the Value-at-Risk optimization problem, which is generally not tractable when the space is continuous. In contrast, our method is grounded in the CRO framework and derives final decisions by directly optimizing over the space of prediction sets, thereby maintaining applicability to continuous decision spaces.

To evaluate the performance of the RAC and CRC methods in regression tasks, we have to make certain adjustments to the RAC method. Specifically, a simple regression problem can be converted into a classification problem via discretization—that is, by partitioning the response and decision space into discrete bins, thus allowing RAC to be applied. However, it is important to note that in general regression settings involving high-dimensional response (such as the 15-dimensional U.S. stock problem in Section 5.2 ), discretization often leads to substantial computational overhead and considerable information loss, making the application of RAC infeasible. To ensure the validity of the RAC method, we consider the following simple regression problem with loss function $\phi(y, z) = -y^\top z$ and decision space $\mathcal{Z} = \{z \in [0,1]^2 : \|z\|_1 = 1\}$. The data is generated by

$$Y_1 = -1.33 \cdot \epsilon_1$$
$$Y_2 = -1 + 0.5 \cdot \epsilon_2$$

where $Y = (Y_1, Y_2) \in \mathbb{R}^2$, and $\epsilon = (\epsilon_1, \epsilon_2) \in \mathbb{R}^2$. The covariate $X \sim N(0, I_2)$ is the spurious feature and noise $\epsilon_1, \epsilon_2$ are independent standard Gaussian random variables. When implementing the RAC method, we need to discretize both the decision space and the label space as follows.

- For the decision space $\mathcal{Z}$, we divide the first dimension $z_1 \in [0, 1]$ into $J$ equally-spaced points $\{z_{1,1}, \ldots, z_{1,J}\}$. Due to the constraint $z_1 + z_2 = 1$, the dicision space is discretized into the finite set $\mathcal{Z}_{\mathrm{dis}} = \{(z_{1,1}, 1 - z_{1,1}), \ldots, (z_{1,J}, 1 - z_{1,J})\}$.
- For the label space $\mathcal{Y}$, we first restrict each dimension of $Y$ to the interval between its $1\%$ and $99\%$ quantiles. This creates a bounded two-dimensional box, which benefits the RAC method by ensuring a bounded loss. This box is then divided uniformly into $L \times L$ regions, and the discretized label space $\mathcal{Y}_{\mathrm{dis}}$ is composed of the top-right endpoints of these regions.

The experimental results are reported in Table 4.

Table 4: The simulation results of CRC and RAC at the nominal level $\alpha = 0.1$, where the sample size is $n = 2000$. For the abbreviation RAC$(J, L)$, the numbers $J, L$ refer to the discretization refinement of decision space and label space, respectively.

| Method | Risk Certificate | Decision Loss | Robustness (%) | Coverage (%) |
|---|---|---|---|---|
| CRC-E | **1.384 ± 0.049** | **0.541 ± 0.065** | **90.0 ± 0.8** | 36.1 ± 1.7 |
| RAC$(6, 2)$ | 1.730 ± 0.039 | 0.803 ± 0.021 | 89.0 ± 1.0 | 89.8 ± 1.1 |
| RAC$(6, 4)$ | 1.687 ± 0.032 | 0.612 ± 0.072 | 90.7 ± 0.9 | 89.9 ± 1.0 |
| RAC$(6, 8)$ | 1.592 ± 0.038 | 0.725 ± 0.042 | 91.6 ± 0.9 | 89.8 ± 1.0 |
| RAC$(11, 2)$ | 1.732 ± 0.038 | 0.803 ± 0.021 | 89.1 ± 1.0 | 89.8 ± 1.0 |
| RAC$(11, 4)$ | 1.684 ± 0.033 | 0.576 ± 0.066 | 91.0 ± 0.9 | 89.9 ± 0.9 |
| RAC$(11, 8)$ | 1.583 ± 0.036 | 0.697 ± 0.042 | 91.6 ± 0.9 | 89.9 ± 1.0 |

### E.5 SIMULATION RESULTS ON CAL-CRC

The experiment results of Cal-CRC under ellipsoid prediction set are shown in Figure 7, where the simulation setting is the same as that in Appendix E.4.

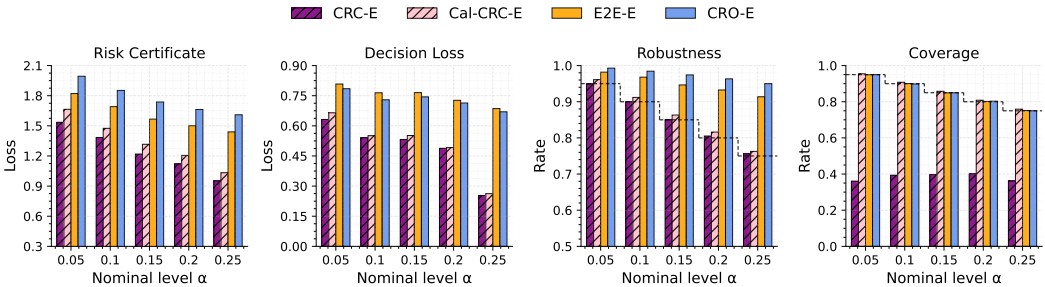

Figure 7: The results of risk certificate, decision loss, robustness, and coverage on synthetic data when varying nominal level $\alpha$ with identical sample size $n = 2000$. The horizontal gray dashed lines refer to robustness levels. The prediction sets are ellipsoids.

### E.6 SIMULATION RESULTS ON POLYHEDRAL PREDICTION SET

In this section, we adopt the methodology from Bärmann et al. (2016) to construct a parametric formulation for polyhedral prediction sets and integrate it into our proposed CRC framework. Simulation experiments demonstrate that under polyhedral prediction sets, our method still exhibits better performance compared to baseline approaches.

Following Bärmann et al. (2016), the derivation of a parametric form for polyhedral prediction sets is inspired by the parametric representation of ellipsoidal prediction sets. Let $\mathbb{B}^q = \{y \in \mathbb{R}^q : \|y\|_2 \leq 1\}$ denote the unit sphere in $\mathbb{R}^q$, and let $\mu_\theta(\cdot) : \mathbb{R}^p \to \mathbb{R}^q$ and $\Sigma_\theta(\cdot) : \mathbb{R}^p \to \mathbb{R}^{q \times q}$ represent the parameterized mean and covariance functions with parameters $\theta$, respectively. The parametric ellipsoidal prediction set can be equivalently defined as:

$$\mathcal{U}_\theta^{\mathrm{E}}(x) = \left\{ y \in \mathbb{R}^q : \Sigma_\theta^{-1/2}(x)(y - \mu_\theta(x)) \in \mathbb{B}^q \right\}.$$

Now, let $\mathcal{B}^q$ be a polyhedral outer $\epsilon$-approximation of $\mathbb{B}^q$, defined by

$$\mathcal{B}^q = \{y : Ky \le k\}, \tag{24}$$

where $K \in \mathbb{R}^{m \times q}, k \in \mathbb{R}^m$ are a fixed matrix and vector, respectively, and $m$ denotes the number of polyhedral facets. The corresponding parametric polyhedral prediction set is then given by:

$$\begin{aligned}
\mathcal{U}_\theta^{\mathrm{P}}(x) &= \left\{ y \in \mathbb{R}^q : \Sigma_\theta^{-1/2}(x)\,(y - \mu(x)) \in \mathcal{B}^q \right\} \\
&= \left\{ y \in \mathbb{R}^q : K\Sigma_\theta^{-1/2}(x)y \le k + K\Sigma_\theta^{-1/2}(x)\mu_\theta(x) \right\}.
\end{aligned} \tag{25}$$

The construction of $\mathcal{B}^q$ depends on the dimension $q$ and the approximation tolerance $\epsilon$. For instance, when $q = 2$ and $\epsilon = 0.01$, an $m = 23$-facet polyhedron ensures that the approximation error remains below $\epsilon$. In this case, the components in (24) are specified as:

$$k = \mathbf{1}_{23}, \quad K = \begin{bmatrix} a_1 \\ a_2 \\ \vdots \\ a_{23} \end{bmatrix} \text{ where } a_i = \left[ \cos\left(\frac{2\pi i}{23}\right), \sin\left(\frac{2\pi i}{23}\right) \right] \text{ for } i = 1, ..., 23.$$

The polyhedral prediction set $\mathcal{U}_\theta^{\mathrm{P}}$ can be directly incorporated into the CRC framework. For example, in a portfolio optimization problem with loss function $\phi(y, z) = -y^\top z$ and decision space $\mathcal{Z} = \{z : z \in [0,1]^q : \|z\|_1 = 1, z \ge 0\}$. We can establish that both the decision $z_\theta(x)$ and the risk certificate $r_\theta(x)$ are differentiable with respect to $\theta$. This enables the search of optimal prediction sets via gradient-based optimization. Furthermore, the theoretical conditions outlined in Section 3.3 continue to hold, ensuring the validity of the corresponding theorems in this extended setting.

In this simulation, we compared the performance of CRC with other methods. The experimental setup remains consistent with that described in Appendix E.4. The experimental results are summarized in Table 5.

Table 5: The simulation results under polyhedral prediction set with nominal level $\alpha = 0.1$, where the sample size is $n = 2000$.

| Method | Risk Certificate | Decision Loss | Robustness (%) | Coverage (%) |
|--------|------------------|---------------|----------------|--------------|
| CRC-P | **1.493** $\pm$ **0.067** | **0.852** $\pm$ **0.037** | **90.3** $\pm$ 0.5 | 23.5 $\pm$ 1.8 |
| CRO-P | 1.844 $\pm$ 0.028 | 0.978 $\pm$ 0.012 | 95.9 $\pm$ 0.6 | 90.1 $\pm$ 0.9 |
| E2E-P | 1.689 $\pm$ 0.024 | 0.954 $\pm$ 0.026 | 93.3 $\pm$ 0.9 | 89.9 $\pm$ 0.9 |

### E.7 ABLATION EXPERIMENTAL RESULTS ON CRC

In this section, we conducted ablation experiments on CRC to compare the performance of CRC and calibrated method. For CRC, we used the parameters of the pre-trained model as the initial values for iteration. For Cal-CRC, we calibrated the model after CRC optimization. For Cal method, we calibrated the pre-trained model. The experimental setup is the same as Section 5.1 and the experimental results are summarized in Table 6.

Table 6: The results of CRC ablation experiments with nominal level $\alpha = 0.1$, where the sample size is $n = 1500$. The prediction sets are ellipsoids.

| Method | Risk Certificate | Decision Loss | Robustness (%) | Coverage (%) |
|--------|------------------|---------------|----------------|--------------|
| CRC-E | **8.660** $\pm$ **0.561** | 7.053 $\pm$ 0.465 | 89.5 $\pm$ 0.8 | 59.8 $\pm$ 5.9 |
| Cal-CRC-E | 9.413 $\pm$ 0.516 | 7.108 $\pm$ 0.469 | **90.9** $\pm$ **1.1** | **90.5** $\pm$ **1.2** |
| Cal-E | 9.581 $\pm$ 0.589 | 7.081 $\pm$ 0.462 | 93.5 $\pm$ 1.3 | 93.2 $\pm$ 1.4 |

