# OpenReview forum: "Conformal Robustness Control: A New Strategy for Robust Decision"
_ICLR.cc/2026/Conference — ICLR 2026 Oral_

### Official Review · Reviewer_HPyM · 2025-10-19

**Soundness:** 4
**Presentation:** 4
**Contribution:** 4
**Rating:** 8
**Confidence:** 4

**Summary:**

There have been a number of works proposing to leverage conformal prediction to do robust decision-making in the context of two-stage decision-making, where an upstream model will specify the parameters of a downstream problem. In the nominal case, for a true problem $\min_w f(w, c)$, a predictor will specify a surrogate parameter estimate $\widehat{c} := g(x)$ against which the optimization is performed as $\min_w f(w, \widehat{c})$. Instead of this, a conformalized approach will produce prediction regions $\mathcal{C}(x)$ such that $\mathcal{P}(C\in\mathcal{C}(x)) \ge 1-\alpha$, from which the robust problem $\min_{w} \max_{\widehat{c}\in\mathcal{C}(x)} f(w, \widehat{c})$ has guarantees of being a valid upper bound on the true optimal value. However, many of these initial works proposed this approach from the perspective that the predictor had been separately conformalized only to be later leveraged in this downstream decision-making task, likely as one of many uses of the conformalized predictor. If, however, the goal is to specifically formulate a robust optimization problem that lends itself to creating an informative, probabilistically valid upper bound, this two-step procedure can be unnecessarily conservative. This paper, therefore, proposes a formulation that directly seeks to define a maximally informative valid upper bound and demonstrates the empirical benefits of doing so compared to two-stage approaches.

**Strengths:**

This paper does a great job identifying a gap in the existing conformal decision-making literature: the idea to directly aim to optimize for the upper bound is very worthwhile. The paper is also very clearly written and the notational cleanly presented, making the insights straightforward to glean from reading. The experimental results were also very clearly presented, demonstrating how the approach produces the desired valid upper bounds while sacrificing coverage, as was the proposed thesis of how the approach was formulated.

**Weaknesses:**

The paper has a couple minor weaknesses that could be improved. First, the theoretical results seem like they would likely be vacuous in many scenarios: Theorem 3.1 has a professed lower bound of $1-\alpha-\Delta_n$, but $\Delta_n$ seems like it would likely be much greater than 1 in any (non-asymptotic) regime. The covering number or Lipschitz constants would likely push it above that point, making this statement not particularly informative. Nonetheless, this statement does have the benefit of providing an asymptotic intuition, which is nice. Overall, this is not a big deal, since the main contributions to me are the insights of directly targeting the prediction region for producing an informative upper bound and the empirical validation more so than the theoretical guarantees. Also, not so much a weakness, but an interesting extension would be to consider prediction regions that are not necessarily just box or ellipsoids but instead unions of such regions (which can still be formulated as a convex program). It would be interesting to know how robustly these findings translate to such prediction regions.

**Questions:**

The paper is a strong contribution with a core insight that is clearly communicated. There are some minor improvements that could likely be made to the theory, but those are minor compared to the overall paper.

---

> ### Author Response · Authors · 2025-11-23
> **Response to Reviewer HPyM**
>
> > **Weaknesses:** The paper has a couple minor weaknesses that could be improved. First, the theoretical results seem like they would likely be vacuous in many scenarios: Theorem 3.1 has a professed lower bound of , but  seems like it would likely be much greater than 1 in any (non-asymptotic) regime. The covering number or Lipschitz constants would likely push it above that point, making this statement not particularly informative. Nonetheless, this statement does have the benefit of providing an asymptotic intuition, which is nice. Overall, this is not a big deal, since the main contributions to me are the insights of directly targeting the prediction region for producing an informative upper bound and the empirical validation more so than the theoretical guarantees. Also, not so much a weakness, but an interesting extension would be to consider prediction regions that are not necessarily just box or ellipsoids but instead unions of such regions (which can still be formulated as a convex program). It would be interesting to know how robustly these findings translate to such prediction regions.
>
> **Response:** We would like to thank you for acknowledging our work's contribution.
>
> (1) The theoretical results are established based on the classical learning theory, and the covering number and Lipschitz conditions are needed to build the non-asymptotic bounds. Similar conditions also appear in Wang et al., (2023).
>
> (2) Our framework also works for the prediction set formed as the union of box or ellipsoid regions. Formally, let $U_{\theta} =  \cup_{k=1}^{K}U_{\theta_{k}}$, where $\theta = (\theta_{1}, ..., \theta_{K})$collects all parameters.
> In this case, if the conditional robust optimization problem under the prediction set $U_{\theta}$ can be reformulated as a convex program, and the implicit gradient method remains applicable for computing the gradients of the risk certificate $r_\theta$ and decision $z_\theta$ with respect to $\theta$. Therefore, the core computational procedure in Algorithm 1 remains fully valid for such composite uncertainty sets. More efficient algorithms to solve such problem need to be further investigated.

---

> ### Comment · Reviewer_HPyM · 2025-11-24
>
> I thank the authors for their rebuttal. Regarding the two points:
>
> 1. I recognize that this is the standard approach to performing such analyses; nonetheless, most PAC/learning theoretic analyses are about asymptotic behavior, in which case the magnitude of the covering numbers is not especially detrimental. In contrast, conformal prediction is generally interested in *finite*-sample coverage, which is why I was pointing out the mismatch in utility in this setting. As stated before, however, I don't think this particular result is central to the core contribution of the rest of the paper.
>
> 2. Yes, I agree with this assessment; I am curious to see how well this holds up empirically.
>
> All of my questions have been answered, and I stand by my score.

---

### Official Review · Reviewer_jy9S · 2025-10-30

**Soundness:** 3
**Presentation:** 4
**Contribution:** 3
**Rating:** 6
**Confidence:** 4

**Summary:**

The paper proposes a prediction set construction method that directly provides robustness in downstream decision-making, thereby bypassing the need for coverage guarantees. The authors present a differentiable loss function to train a predictive model to output parameters for a prediction set with asymptotic robustness guarantees. To provide finite-sample guarantees, they calibrate the prediction sets using conformal prediction. This two-step process is empirically validated on US stock and Battery Storage data, along with synthetic data generated for portfolio optimization.

**Strengths:**

The paper is well-written and organized. The theoretical contributions are meaningful in validating the proposed method. The experiments use a suite of relevant evaluation metrics that further strengthen the paper.

**Weaknesses:**

This paper shares the same motivations as Kiyani et al (2025) which is why I think CRC and Kiyani et al (2025)’s method Risk Averse Calibration (RAC) belong to the same line of research. But I found the explanation of RAC deficient in this paper. Specifically, highlighting and explaining the weaknesses of their method is quite important in helping the reader juxtapose CRC with RAC. This comparison is especially difficult to do since RAC was not included as a baseline in experiments.

If the authors can explain the weaknesses of RAC in detail and compare the empirical performance of RAC with CRC (treat RAC like another baseline), I would be willing to increase my score.

_References_
* Kiyani et al. (2025), Decision theoretic foundations for conformal prediction: Optimal uncertainty quantification for risk-averse agents. https://arxiv.org/pdf/2502.02561.

**Questions:**

* It’s clear that the calibration procedure is necessary to obtain finite sample guarantees, but can we just perform this calibration procedure on any model that outputs point estimates? What is the benefit of using this specific training procedure before calibration? Were there any ablation studies to support this? Similarly, were there any ablations that highlighted the usefulness of the calibration procedure?
* It’s not immediately clear why coverage is included as a performance measure (I understand it’s used to show that coverage isn’t a necessary condition). It would help if there were some foreshadowing on how this will be used.

---

> ### Author Response · Authors · 2025-11-23
> **Response to Reviewer jy9S (Part I)**
>
> > **Weakness.** This paper shares the same motivations as Kiyani et al (2025) which is why I think CRC and Kiyani et al (2025)’s method Risk Averse Calibration (RAC) belong to the same line of research. But I found the explanation of RAC deficient in this paper. Specifically, highlighting and explaining the weaknesses of their method is quite important in helping the reader juxtapose CRC with RAC. This comparison is especially difficult to do since RAC was not included as a baseline in experiments.
> If the authors can explain the weaknesses of RAC in detail and compare the empirical performance of RAC with CRC (treat RAC like another baseline), I would be willing to increase my score.
>
> **Response:**  We thank the reviewer for this constructive feedback and the opportunity to clarify the relationship between our work and Kiyani et al. (2025). We agree that a detailed comparison is crucial, and we have significantly expanded our discussion in the revised manuscript to address this point.
>
> First, we want to highlight that the primary optimization problems underlying CRC (our Eq. 4 in the revision) and the RA-DPO/RA-CPO framework are equivalent, a theorem we have now rigorously proven in Appendix B.2 of our revision. This theoretical equivalence means that, at a conceptual level, both frameworks return the same decision and risk certificate.
>
> The critical distinction lies in implementation and applicability:
>
> -The RAC method, as derived from RA-DPO/RA-CPO, is inherently designed for the classification problem with a finite decision space. It relies on solving a VaR optimization problem to construct the prediction set (ref. Proposition 3.1, Eq. 9-10, Kiyani et al., 2025), which becomes computationally intractable in continuous domains.
>
> -In contrast, our CRC implementation leverages the implicit gradient method, making it naturally suited for and computationally tractable in continuous decision spaces.
>
> We have now included a direct empirical comparison in the continuous case. To implement the RAC method, we discretized the label and decision spaces. The results, presented in the following table, consistently demonstrate the superiority of the CRC framework in continuous settings. Experiment details are given in Appendix E.4.
>
> **Table:** The simulation results of CRC and RAC at the nominal level ($\alpha = 0.1$). For the abbreviation RAC(J, L), the numbers (J, L) refer to the discretization refinement of decision space and label space, respectively.
>
> | Method | Risk Certificate | Decision Loss | Robustness (%) | Coverage (%) |
> | --- | --- | --- | --- | --- |
> | CRC-E | **1.384 ± 0.049** | **0.541 ± 0.065** | **90.0 ± 0.8** | 36.1 ± 1.7 |
> | RAC(6, 2) | 1.730 ± 0.039 | 0.803 ± 0.021 | 89.0 ± 1.0 | 89.8 ± 1.1 |
> | RAC(6, 4) | 1.687 ± 0.032 | 0.612 ± 0.072 | 90.7 ± 0.9 | 89.9 ± 1.0 |
> | RAC(6, 8) | 1.592 ± 0.038 | 0.725 ± 0.042 | 91.6 ± 0.9 | 89.8 ± 1.0 |
> | RAC(11, 2) | 1.732 ± 0.038 | 0.803 ± 0.021 | 89.1 ± 1.0 | 89.8 ± 1.0 |
> | RAC(11, 4) | 1.684 ± 0.033 | 0.576 ± 0.066 | 91.0 ± 0.9 | 89.9 ± 0.9 |
> | RAC(11, 8) | 1.583 ± 0.036 | 0.697 ± 0.042 | 91.6 ± 0.9 | 89.9 ± 1.0 |

---

> > ### Author Response · Authors · 2025-11-23
> > **Response to Reviewer jy9S (Part II)**
> >
> > > **Q1.** It’s clear that the calibration procedure is necessary to obtain finite sample guarantees, but can we just perform this calibration procedure on any model that outputs point estimates? What is the benefit of using this specific training procedure before calibration? Were there any ablation studies to support this? Similarly, were there any ablations that highlighted the usefulness of the calibration procedure?
> >
> > **Response:**  Thank you for this insightful question regarding the role of calibration and our proposed training procedure. The calibration procedure can, in principle, be applied to any pre-trained model to obtain finite-sample coverage guarantees, much like in split conformal prediction. However, if the base model is trained independently of the downstream robust decision task, the resulting decision can be overly conservative.
> >
> > The key benefit of our CRC training is that it explicitly optimizes the model parameters for the risk certificate value of robust decision-making before calibration. As established in Theorems 3.1 and 3.2, this yields a model that already minimizes the risk certificate. The subsequent calibration step then provides the formal finite-sample guarantee on top of this already-optimized solution.
> >
> > To quantitatively validate this advantage, we have included an ablation study comparing our methods (CRC and Cal-CRC) against a baseline (Cal-Pretrain)  where calibration is applied to a standard pre-trained model. The results in the following table show that the risk certificate value of Cal-CRC is still lower than that of Cal-Pretrain, even though the improvement is not significant as CRC.
> >
> > **Table:** The results of CRC ablation experiments under ellipsoid sets with nominal level \(\alpha = 0.1\), where the sample size is \(n = 1500\).
> >
> > | Method    | Risk Certificate   | Decision Loss    | Robustness (%) | Coverage (%) |
> > |-----------|--------------------|------------------|----------------|--------------|
> > | CRC    | **8.660 ± 0.561**  | 7.053 ± 0.465    | 89.5 ± 0.8     | 59.8 ± 5.9   |
> > | Cal-CRC | 9.413 ± 0.516      | 7.108 ± 0.469    | **90.9 ± 1.1** | **90.5 ± 1.2** |
> > | Cal-Pretrain     | 9.581 ± 0.589      | 7.081 ± 0.462    | 93.5 ± 1.3     | 93.2 ± 1.4   |
> >
> > > **Q2.** It’s not immediately clear why coverage is included as a performance measure (I understand it’s used to show that coverage isn’t a necessary condition). It would help if there were some foreshadowing on how this will be used.
> >
> > **Response:**  Thank you for the question. We included the coverage in the experiment results to show that the robustness can be satisfied even though the coverage is out of control. And the coverage is not regarded as a performance measure. We have made this point clear in the revision.

---

> > > ### Comment · Reviewer_jy9S · 2025-11-23
> > >
> > > Thank you for addressing the points I raised. I have revised my score to reflect your updates.

---

> > > > ### Author Response · Authors · 2025-11-24
> > > >
> > > > We sincerely appreciate your decision to raise the score. We are pleased to see that our revisions have successfully addressed your core concerns.

---

### Official Review · Reviewer_KBcd · 2025-11-01

**Soundness:** 3
**Presentation:** 3
**Contribution:** 3
**Rating:** 6
**Confidence:** 3

**Summary:**

This paper proposes a new methodology for the contextual robust optimization problem, where the goal is to optimize the expectation of some worst-case performance for a high probability robust set. To solve this problem, a typical method is to first construct this robust set and then optimize. Instead, in this paper, the authors proposed a new method of combining robust set construction and objective optimization together, which enjoys a better performance. The authors also use theoretical and numerical results to illustrate the strength of the algorithm.

**Strengths:**

1. The setup, methodology, and results are very clear.
2. The methodology is novel, and the results look solid.
3. Both finite-sample theoretical results and numerical illustrations demonstrate the strength of the new methodology.

**Weaknesses:**

1. The paper mainly focuses on box/ellipsoid robust sets. Is this algorithm also compatible with other forms of robust sets?

2. Theorem is still an overall coverage statement. I wonder whether the authors could show a conditional statement, like with high probability, P(\phi(y,z(X))\leq r(X)|X)\geq 1-alpha-Delta

**Questions:**

In addition to the question in the weaknesses section,

In Algorithm 1, what would be a good number of gradient descent steps?

---

> ### Author Response · Authors · 2025-11-23
> **Response to Reviewer KBcd**
>
> > **W1.** The paper mainly focuses on box/ellipsoid robust sets. Is this algorithm also compatible with other forms of robust sets?
>
> **Response:**  We thank the reviewer for this insightful question.
>
> Our CRC framework is indeed compatible with general forms of prediction sets beyond boxes and ellipsoids, provided they can be suitably parameterized. The theoretical guarantees in Section 3 apply to general convex sets, and the optimization procedure can accommodate any parametrization that allows efficient projection or constraint handling.
> To empirically demonstrate this flexibility, we have added a new experiment utilizing polyhedral uncertainty sets from Bärmann et al. (2016). The results, presented in the following table, show that our method maintains strong performance under this more complex uncertainty geometry. Detailed setup and further discussion can be found in Appendix E.6 (revised version) of the revised manuscript.
>
> **Table: Experiment results under polyhedral set with nominal level ($\alpha = 0.1$)**
>
> | Method | Risk Certificate | Decision Loss | Robustness (%) | Coverage (%) |
> | --- | --- | --- | --- | --- |
> | CRC-P | **1.493 ± 0.067** | **0.852 ± 0.037** | **90.3 ± 0.5** | 23.5 ± 1.8 |
> | CRO-P | 1.844 ± 0.028 | 0.978 ± 0.012 | 95.9 ± 0.6 | 90.1 ± 0.9 |
> | E2E-P | 1.689 ± 0.024 | 0.954 ± 0.026 | 93.3 ± 0.9 | 89.9 ± 0.9 |
>
> [1] Bärmann, A., Heidt, A., Martin, A., Pokutta, S., & Thurner, C. (2016). Polyhedral approximation of ellipsoidal uncertainty sets via extended formulations: a computational case study. Computational Management Science, 13(2), 151-193.
>
> > **W2.** Theorem is still an overall coverage statement. I wonder whether the authors could show a conditional statement, like with high probability, P(\phi(y,z(X))\leq r(X)|X)\geq 1-alpha-Delta.
>
> **Response:**  Thank you for the constructive comment. To achieve such a conditional robustness guarantee, the  constraint in the empirical optimization problem (5) should be replaced to approximate the conditional probability. We can use the locally weighted method in nonparametric statistics for approximation. Let $H(.,.)$ be a kernel, and define the weights $w_i(X) = H(X_i, X)/\sum_{j=1}^n H(X_j, X)$for $i=1,...,n$. Then we consider the following constrained problem:
> $$
> \min_{\theta \in \Theta} r_{\theta}(X_i)\ s.t.\ \sum_{i=1}^n w_i(X) 1\{\phi(Y_i, z_{\theta}(X_i)) \leq r_{\theta}(X_i)\} \geq 1-\alpha.
> $$
> The corresponding theoretical guarantee still needs to be further explored.
>
> > **Q.** In addition to the question in the weaknesses section, In Algorithm 1, what would be a good number of gradient descent steps?
>
> **Response:**  We thank the reviewer for raising this important implementation detail. Since the time complexity of convergence for such optimization problems has not been analyzed in prior work e.g., Davis et al. (2020) and Bolte et al. (2021), we cannot choose a good number of gradient steps. In our implementation, early stopping is triggered when the loss on the validation set fails to decrease for 10 consecutive iterations or when the predefined maximum number of iterations is reached.

---

### Official Review · Reviewer_Cx82 · 2025-11-01

**Soundness:** 3
**Presentation:** 2
**Contribution:** 3
**Rating:** 6
**Confidence:** 5

**Summary:**

This paper proposes Conformal Robustness Control (CRC), a framework for robust decision making that directly enforces a robustness constraint on the loss, rather than the traditional coverage constraint used in Conditional Robust Optimization (CRO). CRC optimizes parameterized prediction sets so that the probability the actual loss exceeds a risk certificate is below a target threshold. The authors provide a differentiable algorithm, non asymptotic theoretical guarantees, and a test time calibration procedure.

**Strengths:**

1. CRC is formulated as a constrained optimization problem with a direct robustness constraint, and solved using a smoothed dual approach suitable for gradient-based learning.
2. The paper provides non asymptotic bounds on robustness and optimality, and extends these to finite sample guarantees via test time calibration.
3. The algorithms and theoretical results are clearly described, and the motivation is well articulated.

**Weaknesses:**

1. The paper refers to CRO as "Contextual Robust Optimization" whereas the foundational literature (e.g., Chenreddy et al., 2022) defines CRO as "Conditional Robust Optimization". This is not just a semantic issue, the "conditional" aspect is central to the theoretical guarantees and the structure of the uncertainty sets. The paper does not clarify this distinction, which may confuse readers and obscure the technical lineage of the framework.
2. The paper claims equivalence between CRC and RA-DPO/RA-CPO (Kiyani et al., 2025) in regression settings, but does not provide a rigorous theoretical or empirical comparison of solution quality, tractability, or optimality gaps in more general settings (e.g., non-convex or discrete decision spaces).
3.  The robustness constraint is enforced via a smoothed indicator (error function), which introduces bias. Theoretical analysis of the impact of the smoothing parameter on constraint satisfaction and optimality is limited, and there is no discussion of feasibility restoration or dual variable stability.
4.  While CRC can achieve robustness with lower coverage, the theoretical implications of this tradeoff are not fully explored. In some applications, coverage may be required for interpretability or regulatory reasons, and the paper does not analyze the joint feasibility or Pareto frontier of robustness and coverage constraints.
5. Theoretical results and experiments are limited to box and ellipsoidal prediction sets. It is unclear how the CRC framework extends to more complex uncertainty sets (e.g., polyhedral, Wasserstein balls, or discrete spaces), and whether the theoretical guarantees hold in these cases.

**Questions:**

1. Can you provide a more rigorous theoretical comparison between CRC and RA-DPO/RA-CPO or CVaR-based robust optimization, especially in non-convex or discrete decision spaces? Are there cases where CRC is strictly more efficient or yields better optimality gaps?
2. How does the choice of smoothing parameter in the surrogate constraint affect the theoretical robustness guarantee and optimality gap? Is there a principled way to select this parameter, and can you bound the bias introduced?
3. Is it possible to enforce both robustness and coverage constraints simultaneously in the CRC framework? What are the theoretical tradeoffs or limitations in doing so?
4. How does CRC extend to more general uncertainty sets (e.g., polyhedral, Wasserstein, or discrete sets)? Do the non-asymptotic guarantees still hold, or are there new challenges?
5. Can you provide statistics on the frequency and magnitude of robustness constraint violations during training and after convergence, compared to CRO and end-to-end baselines?
6. How sensitive is CRC to the smoothing parameter, Lagrange multiplier update schedule, and calibration split size? Does constraint satisfaction degrade for certain settings?
7. What is the computational cost and convergence behavior of CRC compared to RA-DPO, CVaR, and end-to-end CRO baselines?

---

> ### Author Response · Authors · 2025-11-23
> **Response to Reviewer Cx82 (Part I)**
>
> > **W1.** The paper refers to CRO as "Contextual Robust Optimization" whereas the foundational literature (e.g., Chenreddy et al., 2022) defines CRO as "Conditional Robust Optimization". This is not just a semantic issue, the "conditional" aspect is central to the theoretical guarantees and the structure of the uncertainty sets. The paper does not clarify this distinction, which may confuse readers and obscure the technical lineage of the framework.
>
> **Response:** We thank the reviewer for this crucial clarification. Our work is indeed based on the "Conditional Robust Optimization" (CRO) framework of Chenreddy et al. (2022). We have corrected this oversight in the entire text. Our formulation follows the cited literature precisely, defining the prediction set as $\mathcal{U}(X)$ depending on the context $X$ and solving the corresponding min-max problem $z(X) = \arg\min_{z\in\mathcal{Z}} \max_{c\in \mathcal{U}(X)} \phi(c,z)$.
>
> > **W2.** The paper claims equivalence between CRC and RA-DPO/RA-CPO (Kiyani et al., 2025) in regression settings, but does not provide a rigorous theoretical or empirical comparison of solution quality, tractability, or optimality gaps in more general settings (e.g., non-convex or discrete decision spaces).
>
> **Response:** We thank the reviewer for this insightful comment regarding the comparison between CRC and RA-DPO/RA-CPO.
> First, for the primary optimization, the CRC problem defined in Eq. (4) (of revision) and the RA-DPO/RA-CPO problem are equivalent. This holds regardless of whether the problem is regression or classification, and whether the decision space is non-convex or discrete. We state this equivalence in Theorem 3.1, and the detailed proof is provided in Appendix B.2 of the revised paper.
> Second, regarding practical implementation, the two approaches indeed differ in their applicability:
>
>  - Methods derived from the RA-DPO/RA-CPO framework, such as RAC, are naturally suited to classification problems with a discrete and finite decision space. The reason is that RAC requires solving the VaR optimization problem, which is generally not tractable for the continuous decision space.
>
> - In contrast, our CRC method is designed for continuous decision spaces. To empirically compare the two methods under this case, we have added a new experiment in which we adapted the RA-CPO approach by discretizing the label and decision spaces. The simulation results are reported in the following table. Experiment details are given in Appendix E.4.
>
> **Table:** The simulation results of CRC and RAC at the nominal level ($\alpha = 0.1$). For the abbreviation RAC(J, L), the numbers (J, L) refer to the discretization refinement of decision space and label space, respectively.
>
> | Method | Risk Certificate | Decision Loss | Robustness (%) | Coverage (%) |
> | --- | --- | --- | --- | --- |
> | CRC-E | **1.384 ± 0.049** | **0.541 ± 0.065** | **90.0 ± 0.8** | 36.1 ± 1.7 |
> | RAC(6, 2) | 1.730 ± 0.039 | 0.803 ± 0.021 | 89.0 ± 1.0 | 89.8 ± 1.1 |
> | RAC(6, 4) | 1.687 ± 0.032 | 0.612 ± 0.072 | 90.7 ± 0.9 | 89.9 ± 1.0 |
> | RAC(6, 8) | 1.592 ± 0.038 | 0.725 ± 0.042 | 91.6 ± 0.9 | 89.8 ± 1.0 |
> | RAC(11, 2) | 1.732 ± 0.038 | 0.803 ± 0.021 | 89.1 ± 1.0 | 89.8 ± 1.0 |
> | RAC(11, 4) | 1.684 ± 0.033 | 0.576 ± 0.066 | 91.0 ± 0.9 | 89.9 ± 0.9 |
> | RAC(11, 8) | 1.583 ± 0.036 | 0.697 ± 0.042 | 91.6 ± 0.9 | 89.9 ± 1.0 |
>
> We acknowledge that our current CRC implementation relies on implicit gradient methods, which are not directly applicable in discrete decision spaces. In such cases, we would recommend using the prediction set construction method proposed by Kiyani et al. (2025). To ensure a balanced and clear discussion, we have added a detailed comparison of both works at the end of Section 3.1 in the revised paper, clarifying the theoretical connections and practical distinctions.
>
> > **W3.** The robustness constraint is enforced via a smoothed indicator (error function), which introduces bias. Theoretical analysis of the impact of the smoothing parameter on constraint satisfaction and optimality is limited, and there is no discussion of feasibility restoration or dual variable stability.
>
> **Response:** The theoretical analysis of the impact of the smoothing parameter is provided in **Appendix D.2** of our submitted version. The optimal choice of smoothing parameter is $\sigma = O(n^{-1/2})$, then the corresponding bounds on robustness constraint and optimality can match those in Theorems 3.2 and 3.3 for the original optimization problems.  This ensures that the smoothed constraint closely approximates the original non-smoothed constraint while maintaining computational tractability.
> Regarding feasibility restoration and dual variable stability, we agree that these are valuable considerations from an optimization perspective. A rigorous analysis of these properties would require additional technical developments beyond the current scope of our work, and we will leave it as future work.

---

> ### Author Response · Authors · 2025-11-23
> **Response to Reviewer Cx82 (Part II)**
>
> > **W4.** While CRC can achieve robustness with lower coverage, the theoretical implications of this tradeoff are not fully explored. In some applications, coverage may be required for interpretability or regulatory reasons, and the paper does not analyze the joint feasibility or Pareto frontier of robustness and coverage constraints.
>
> **Response:** Thank you for the comment. For some applications, coverage may be required for interpretability or regulatory compliance. In our work, however, we focus on settings where robustness is the primary design objective. As discussed in the paper, coverage is a sufficient but not necessary condition for robustness. This means that if both constraints were included in the same formulation, the coverage requirement would dominate the feasible region, potentially leading to overly conservative decisions when the true goal is robustness. Imposing both constraints would not yield a Pareto-type tradeoff in this context, but rather collapse the problem to a coverage-only formulation. Our framework is therefore designed to directly control robustness, which provides a more efficient and targeted approach for applications where robustness itself is the key operational requirement.
>
> > **W5.** Theoretical results and experiments are limited to box and ellipsoidal prediction sets. It is unclear how the CRC framework extends to more complex uncertainty sets (e.g., polyhedral, Wasserstein balls, or discrete spaces), and whether the theoretical guarantees hold in these cases.
>
> **Response:** We thank the reviewer for this valuable comment regarding the generality of the CRC framework.
>
> - We would like to clarify that our theoretical results in Section 3 are established for general prediction sets and are not limited to specific geometries. The framework can indeed accommodate polyhedral and other convex uncertainty sets, as demonstrated by our new experiment incorporating the polyhedral sets from Bärmann et al. (2016), with results included in the following table. Our method still performs better than other baselines. The detailed setting is given in **Appendix E.6** of the revision.
>
> **Table:** Experiment results under the polyhedral set with nominal level ($\alpha = 0.1$)
>
> | Method | Risk Certificate | Decision Loss | Robustness (%) | Coverage (%) |
> | --- | --- | --- | --- | --- |
> | CRC-P | **1.493 ± 0.067** | **0.852 ± 0.037** | **90.3 ± 0.5** | 23.5 ± 1.8 |
> | CRO-P | 1.844 ± 0.028 | 0.978 ± 0.012 | 95.9 ± 0.6 | 90.1 ± 0.9 |
> | E2E-P | 1.689 ± 0.024 | 0.954 ± 0.026 | 93.3 ± 0.9 | 89.9 ± 0.9 |
>
> - Regarding computational aspects, the resulting optimization problem can be solved by Algorithm 1 if the decision space is continuous and the implicit differential can be applied.
>
> - For Wasserstein balls, we note they primarily serve as ambiguity sets in distributionally robust optimization (DRO) to model distributional uncertainty, whereas our CRO framework constructs uncertainty sets for realizations of the random parameters. These are conceptually distinct objectives.
>
> [1] Bärmann, A., Heidt, A., Martin, A., Pokutta, S., & Thurner, C. (2016). Polyhedral approximation of ellipsoidal uncertainty sets via extended formulations: a computational case study. Computational Management Science, 13(2), 151-193.

---

> > ### Author Response · Authors · 2025-11-23
> > **Response to Reviewer Cx82 (Part III)**
> >
> > > **Q1.** Can you provide a more rigorous theoretical comparison between CRC and RA-DPO/RA-CPO or CVaR-based robust optimization, especially in non-convex or discrete decision spaces? Are there cases where CRC is strictly more efficient or yields better optimality gaps?
> >
> > **Response:**  For the equivalence between  RA-DPO/RA-CPO and CRC, please see the response to weakness 1. In Appendix B.3, we show that the parametrized CRC can achieve a lower risk certificate than the parametrized RA-CPO. Regarding CVaR-based robust optimization, a direct comparison of optimality is more nuanced because the two approaches enforce robustness through different constraint structures. While both methods provide probabilistic guarantees, their feasible regions and consequent optimal solutions generally differ. The relative efficiency and optimality gaps between CRC and CVaR-based methods would therefore be problem-dependent.
> >
> > > **Q2** How does the choice of smoothing parameter in the surrogate constraint affect the theoretical robustness guarantee and optimality gap? Is there a principled way to select this parameter, and can you bound the bias introduced?
> >
> > **Response:**  Please see the response to Weakness 3.
> >
> > > **Q3**  Is it possible to enforce both robustness and coverage constraints simultaneously in the CRC framework? What are the theoretical tradeoffs or limitations in doing so?
> >
> > **Response:**  Since coverage is a sufficient condition for robustness in our framework, imposing both constraints would reduce to enforcing the coverage constraint alone. This would defeat our core objective of directly controlling robustness, which is the central motivation behind the CRC framework.
> > By focusing solely on robustness, CRC avoids the potential conservatism of coverage-based approaches and provides a more targeted and often more efficient means of achieving the desired probabilistic guarantees. Imposing both constraints would not yield a Pareto-type tradeoff in this context, but rather collapse the problem to a coverage-only formulation.
> >
> > > **Q4** How does CRC extend to more general uncertainty sets (e.g., polyhedral, Wasserstein, or discrete sets)? Do the non-asymptotic guarantees still hold, or are there new challenges?
> >
> > **Response:**  The non-asymptotic guarantees in Theorems 3.2 and 3.3 still hold for general uncertainty sets as long as Conditions 3.1 and 3.2 are satisfied. Please refer to the response to Weakness 5.
> >
> > > **Q5** Can you provide statistics on the frequency and magnitude of robustness constraint violations during training and after convergence, compared to CRO and end-to-end baselines?
> >
> > **Response:**  Thank you for the question. We have added the statistics during training and after convergence of CRC in the following tables. The experiment setting is the same as Section 5.1. The pre-training violation frequency is 0 because the corresponding prediction set is very conservative, leading to a large risk certificate. "Mid-train" refers to the middle of the training process.
> > Since CRO and end-to-end baselines are not constrained optimization problems, there are no constraint violations.
> >
> > **Table:** Robustness constraint violations in 100 simulations with nominal level ($\alpha=0.1$)
> >
> > | Method | Converge epochs | Violation Frequency |  |  | Violation Magnitude |  |  |
> > | --- | --- | --- | --- | --- | --- | --- | --- |
> > |  |  | Pre-train | Mid-train | Converge | Pre-train | Mid-Train | Converge |
> > | CRC-E  | 70 | 0/100 | 19/100 | 23/100 | 0% | 0.82% | 1.08% |

---

> > > ### Author Response · Authors · 2025-11-23
> > > **Response to Reviewer Cx82 (Part IV)**
> > >
> > > > **Q6** How sensitive is CRC to the smoothing parameter, Lagrange multiplier update schedule, and calibration split size? Does constraint satisfaction degrade for certain settings?
> > >
> > > **Response:**  Thank you for the question. In the following tables, we provide the sensitivity analysis of CRC on the smoothing parameter, Lagrange multiplier update schedule (update $\lambda$ every $\tau$ steps, $\tau \in \{1,2,4,8\}$) under the experiment setting of Section 5.1. The detailed setting is given in Appendix E.1 of the revision.  For the sample size, the results are plotted in Figure 3. We can see that the final performance of CRC is quite stable.
> > >
> > > **Table:** Sensitivity analysis on smoothing parameter with nominal level ($\alpha = 0.1$)
> > >
> > > | Smoothing parameter _σ_ | Risk Certificate | Decision Loss | Robustness (%) | Coverage (%) |
> > > | --- | --- | --- | --- | --- |
> > > | 0.01 | 8.678 ± 0.299 | 7.072 ± 0.220 | 89.8 ± 0.7 | 60.8 ± 5.6 |
> > > | 0.05 | 8.633 ± 0.295 | 7.070 ± 0.219 | 89.9 ± 0.6 | 59.6 ± 5.5 |
> > > | 0.10 | 8.641 ± 0.306 | 7.071 ± 0.221 | 90.3 ± 0.5 | 59.4 ± 5.7 |
> > > | 0.15 | 8.643 ± 0.315 | 7.070 ± 0.221 | 90.5 ± 0.6 | 59.6 ± 5.7 |
> > > | 0.20 | 8.649 ± 0.308 | 7.070 ± 0.220 | 90.2 ± 0.5 | 59.6 ± 5.7 |
> > >
> > > **Table:** Sensitivity analysis on Lagrange multiplier update schedule with nominal level ($\alpha = 0.1$)
> > >
> > > | λ update schedule | Risk Certificate | Decision Loss | Robustness (%) | Coverage (%) |
> > > | --- | --- | --- | --- | --- |
> > > | 1 | 8.641 ± 0.334 | 7.109 ± 0.251 | 89.9 ± 0.5 | 58.7 ± 5.8 |
> > > | 2 | 8.528 ± 0.302 | 7.106 ± 0.251 | 90.4 ± 0.6 | 56.4 ± 5.5 |
> > > | 4 | 8.478 ± 0.278 | 7.105 ± 0.251 | 90.1 ± 0.5 | 55.4 ± 4.8 |
> > > | 8 | 8.452 ± 0.282 | 7.105 ± 0.251 | 89.7 ± 0.6 | 55.2 ± 4.9 |
> > >
> > > > **Q7.** What is the computational cost and convergence behavior of CRC compared to RA-DPO, CVaR, and end-to-end CRO baselines?
> > >
> > > **Response:**  Thank you for the question. For the continuous decision space, RA-DPO is not tractable due to the VaR optimization. In our experiments, we find that the epoch numbers required to converge by the CRC and the end-to-end method are close, see the table in the response to Q5. The CVaR method is not considered in our paper because the robustness constraints are different.

---

### Author Response · Authors · 2025-12-03
**General response**

We thank the reviewers and the AC for handling our paper. We have made substantial revisions to improve clarity, and below summarize three key updates.

1. **Connection and distinction between our work and Kiyani et al. (2025)**
   The core optimization problems behind CRC (our revised Eq. 4) and the RA‑DPO/RA‑CPO framework are equivalent, as stated in Theorem 3.1 of the revised manuscript. The critical difference lies in implementation and applicability:
   - The RAC method derived from RA‑DPO/RA‑CPO is inherently designed for classification with a finite decision space. It requires solving a VaR optimization to construct the prediction set (see Proposition 3.1, Eqs. 9–10 of Kiyani et al., 2025), which becomes computationally intractable in continuous domains.
   - In contrast, our CRC implementation leverages the implicit gradient method, making it naturally suitable and computationally tractable for continuous decision spaces. Following reviewer suggestions, we now include an empirical comparison, implementing RAC by discretizing the label and decision spaces. Please see our responses to Reviewers Cx82 and jy9S for details.

We have updated Section 3.1 to clarify these connections and distinctions.

2. **More complex prediction sets.**
   While the manuscript illustrated our framework with parameterized ellipsoid and box sets, CRC can in principle accommodate arbitrary prediction sets provided the corresponding CRO problem remains convex and implicit differentiation is applicable. The theoretical results in Section 3.3 continue to hold under these conditions. As suggested by reviewers, we have added a new experiment using a polyhedral prediction set; see the response to Reviewers Cx82 and KBcd.

3. **Additional experiments**
   We have included a sensitivity analysis of CRC with respect to the smoothing parameter and the Lagrange multiplier update schedule, demonstrating the stability of our method. An ablation study compares CRC and Cal‑CRC against a calibrated pre‑trained baseline (Cal‑Pretrain), highlighting the benefit of the CRC training process in improving decision efficiency.

We believe the revised manuscript addresses the reviewers’ questions and concerns while improving clarity and completeness. We are grateful for their valuable feedback and hope the final version reflects these improvements.

---

### Meta-Review · Area_Chair_tfM7 · 2026-01-08

**Summary:**

Their were two key concerns raised by reviewers:
1. Claimed equivalence between proposed method and RAC/RA-DPO/RA-CPO (Kiyani et al., 2025) is not supported by sufficient theoretical or empirical evidence (Cx82, jy9S)
2. Theory and experiments limited to box and ellipsoidal prediction sets (Cx82, KBcd, HPyM)

**Reviewer Concerns:**

Both of these concerns have been resolved. The authors addressed the concerns as follows:
1. The authors updated section 3.1 to more precisely characterize the relationship between the proposed method and Kiyani et al. (2025). They also clarified that the proposed method focuses on continuous spaces (as opposed to discrete spaces) and added new experiments comparing the proposed method to RAC (an exemplar of the RA-DPO/RA-CPO framework). These results show that the proposed method offers a tighter risk certificate than RAC.
2. The authors added new experiments using polyhedral sets which shows clear improvement over baselines.

**Reviewer Scores:**

- jy9S is very likely to increase their score. Their review stated that they would be willing to increase their score if the authors could clarify the relationship between the proposed method and RAC along with empirical results. The authors addressed these changes during the rebuttal phase (see concern 1 above).
- HPyM is virtually certain to keep their score. Their review was already very positive and they did not have any serious concerns with the paper.
- Cx82 is about as likely as not to increase their score. The authors addressed concerns about the equivalence between the proposed method and RA-DPO/RA-CPO through both theoretical and empirical arguments. They also added new experiments with polyhedral sets and clarified that there is not a robustness/coverage tradeoff but rather a choice about whether coverage is important or robustness is the true goal. As their initial review was already positive, Cx82 is about equally likely to increase their score as they are to keep their original score.
- KBcd is about as likely as not to increase their score.  The authors addressed concerns about generality by adding new experiments with polyhedral sets and clarified that the analysis could potentially be adapted to the conditional coverage case but that theoretical properties still need to be explored. As their initial review was already positive, KBcd is about equally likely to increase their score as they are to keep their score.

---

### Decision · Program_Chairs · 2026-01-26

Accept (Oral)